# Ectodermal Wnt signaling, cell fate determination, and polarity of the skate gill arch skeleton

Jenaid M Rees[1], Victoria A Sleight[2], Stephen J Clark[3], Tetsuya Nakamura[4], J Andrew Gillis[1,5]*

[1]Department of Zoology, University of Cambridge, Cambridge, United Kingdom; [2]School of Biological Sciences, University of Aberdeen, Aberdeen, United Kingdom; [3]The Babraham Institute, Cambridge, United Kingdom; [4]Department of Genetics, Rutgers University, Piscataway, United States; [5]Josephine Bay Paul Center for Comparative Molecular Biology and Evolution, Marine Biological Laboratory, Woods Hole, United States

*For correspondence:
agillis@mbl.edu

Competing interest: The authors declare that no competing interests exist.

**Abstract** The gill skeleton of cartilaginous fishes (sharks, skates, rays, and holocephalans) exhibits a striking anterior–posterior polarity, with a series of fine appendages called branchial rays projecting from the posterior margin of the gill arch cartilages. We previously demonstrated in the skate (*Leucoraja erinacea*) that branchial rays derive from a posterior domain of pharyngeal arch mesenchyme that is responsive to Sonic hedgehog (Shh) signaling from a distal gill arch epithelial ridge (GAER) signaling centre. However, how branchial ray progenitors are specified exclusively within posterior gill arch mesenchyme is not known. Here, we show that genes encoding several Wnt ligands are expressed in the ectoderm immediately adjacent to the skate GAER, and that these Wnt signals are transduced largely in the anterior arch environment. Using pharmacological manipulation, we show that inhibition of Wnt signalling results in an anterior expansion of Shh signal transduction in developing skate gill arches, and in the formation of ectopic anterior branchial ray cartilages. Our findings demonstrate that ectodermal Wnt signalling contributes to gill arch skeletal polarity in skate by restricting Shh signal transduction and chondrogenesis to the posterior arch environment and highlights the importance of signalling interactions at embryonic tissue boundaries for cell fate determination in vertebrate pharyngeal arches.

## Editor's evaluation

Building on previous work dissecting the polarisation of a gill arch epithelial ridge (GAER) of a cartilaginous fish (skate), this paper uses RNA–Sequencing, multiplexed HCR, cell fate mapping, and pharmacological manipulation to uncover the contribution of the Wnt signalling pathway to the patterning of the gill arch, highlighting the role tissues boundaries in signalling and patterning embryonic tissues.

## Introduction

The pharyngeal arches of vertebrates are a series of paired columns of tissue that form on either side of the embryonic head. Pharyngeal arches form as iterative outpockets of foregut endoderm contact the overlying surface ectoderm, and this meeting of endoderm and ectoderm generates columns lined laterally by ectoderm, medially by endoderm, and containing a core of mesoderm and neural crest-derived mesenchyme (*Graham and Smith, 2001*). In fishes, endodermal outpockets fuse with

**Figure 1.** Overview of skate gill arch skeletal anatomy. Skeletal preparation of an S33 skate embryo in (**A**) ventral and (**B**) lateral view. In (**A'**) and (**B'**) the gill arches and branchial rays are highlighted, with branchial rays false-colored red. Note that branchial rays project from the posterior margin of the hyoid and first four gill arches. (**C**) A dissected gill arch with the (**C'**) dorsal epibranchial and ventral ceratobranchial gill arch cartilages indicated, and with branchial rays false-colored red. a–p indicates anterior–posterior axis. br: branchial ray, cb: ceratobranchial, eb: epibranchial, ga1-4: gill arches 1–4, ha: hyoid arch, ma: mandibular arch. Scale bars: 1 mm.

the overlying surface ectoderm, giving rise to the gill slits and the respiratory surfaces of the gills (*Gillis and Tidswell, 2017*). In amniotes, endodermal outpockets give rise to glandular tissues, such as the tonsils, parathyroid, and ultimobranchial glands (*Grevellec and Tucker, 2010*). The largely neural crest-derived mesenchyme of the pharyngeal arches gives rise to the pharyngeal skeleton – i.e., the skeleton of the jaws and gills in fishes, and of the jaw, auditory ossicles, and larynx in amniotes (*Jiang et al., 2002*; *Kague et al., 2012*; *Sleight and Gillis, 2020*; *Couly and Le Douarin, 1990*) – with mesenchymal derivatives of each arch receiving patterning and polarity information via signals from adjacent epithelia (*Veitch et al., 1999*; *Gillis et al., 2009a*, *Couly et al., 2002*; *Brito et al., 2006*). Pharyngeal arches are named according to their ancestral skeletal derivatives: the 1st pharyngeal arch is termed the mandibular arch, the 2nd pharyngeal arch is the hyoid arch, while the caudal pharyngeal arches are collectively termed the gill arches. In fishes, the gill arches give rise to a series of skeletal 'arches' that support the lamellae of the gills.

All jawed vertebrates belong to one of two lineages: cartilaginous fishes (sharks, skates, rays, and holocephalans) or bony vertebrates (ray- and lobe-finned fishes, with the latter including tetrapods). While the gill arch skeleton of both cartilaginous and bony fishes ancestrally consisted proximally of two principal gill arch cartilages (the epi- and ceratobranchials), the gill arch skeleton of cartilaginous fishes additionally includes a distal series of fine cartilaginous rods called branchial rays (*Figure 1A–C*). These rays reflect the clear anteroposterior polarity of the gill arch skeleton of cartilaginous fishes, originating along the posterior margin of the epi- and ceratobranchial cartilage, and curving posteriorly as they project into the interbranchial septum of each arch (*Gillis et al., 2009b*). Elasmobranch cartilaginous fishes (sharks, skates, and rays) possess five sets of branchial rays, associated with their hyoid and first four gill arches. Holocephalans, on the other hand, possess a single set of branchial rays supporting their hyoid arch-derived operculum (*Gillis et al., 2011*).

The branchial rays of cartilaginous fishes develop under the influence of the GAER (gill arch epithelial ridge): a *Sonic hedgehog* (*Shh*)-expressing signaling centre located within the posterior–distal epithelium of the gill arches (*Gillis et al., 2009b*; *Gillis and Hall, 2016*). As the gill arches undergo a prolonged phase of lateral expansion, Shh signaling from the GAER is asymmetrically transduced within the posterior arch environment, as evidenced by posterior expression of *Ptc2* (*Gillis and Hall, 2016*) – a Shh co-receptor and transcriptional readout of Shh signaling (*Pearse et al., 2001*). This posterior transduction of Shh signaling, in turn, appears to underlie the anteroposterior polarity of the gill arch skeleton: in skate, branchial rays derive exclusively from posterior–distal (Shh-responsive/ GAER-adjacent) gill arch mesenchyme, application of exogenous Shh protein within gill arches is sufficient to induce ectopic branchial ray formation, and targeted or systemic inhibition of hedgehog signaling using cyclopamine results in branchial ray deletion (*Gillis et al., 2009b*; *Gillis and Hall, 2016*). However, whether other signaling mechanisms function alongside or in conjunction with GAER Shh signaling to establish and maintain gill arch skeletal polarity remains unexplored.

Here, we show in the little skate (*L. erinacea*) that the ectoderm immediately adjacent to the GAER expresses several genes encoding Wnt ligands. These Wnt signals are transduced in the anterior arch environment in a pattern that is broadly complementary to the posterior transduction of GAER Shh signaling. Inhibition of Wnt signaling in developing skate embryos results in an anterior expansion of Shh signaling transduction, and in the formation of ectopic branchial rays in the anterior gill arch. We propose that Wnt signaling from the pharyngeal arch ectoderm contributes to the maintenance of anterior–posterior polarity of the skate gill arch skeleton by repressing Shh signaling and chondrogenesis to the posterior gill arch territory.

## Results

### *Wnt* gene expression in the skate GAER and GAER-adjacent epithelium

In developing skate gill arches, *Shh* is expressed in the GAER (*Figure 2A*), and branchial rays develop from a GAER-responsive domain of posterior arch mesenchyme (*Gillis and Hall, 2016*). To further explore the transcriptional environment of the GAER and to discover additional gene expression features that may contribute to gill arch polarity, we conducted a comparative transcriptomic and differential gene expression analysis of GAER and non-GAER regions of the first gill arch of the embryonic skate. Briefly, we manually dissected the (**1**) GAER and (**2**) non-GAER (control) regions from the first gill arch of stage (S)26 skate embryos (n=5; *Figure 2B–Bi*), and we performed RNA extraction, library preparation, and RNAseq analysis on these samples. For the GAER region of the first gill arch, we included the whole distal tip of the arch in our dissection to ensure that the *Shh*-expressing GAER and adjacent Shh-responsive (i.e. *Ptc2*+) tissues were captured for this analysis (*Figure 2C–D*). Following *de novo* transcriptome assembly, we tested for differential gene expression between GAER and non-GAER tissues (after first testing for differential expression of *Shh* in the GAER region, to confirm dissection accuracy). We used a false discovery rate of 5% and no log fold-change (LogFC) cut-off to generate a list of 401 genes that were upregulated in the GAER region. We sorted this list for genes encoding signaling pathway components and transcription factors with a log-fold change >2 (*Figure 2E*). Differentially expressed genes were subjected to functional enrichment analysis, and this analysis indicated enrichment of Wnt signaling pathway components in the GAER region (*Figure 2—figure supplement 1* for enrichment analysis). Differentially expressed genes in the GAER region included those encoding the Wnt ligands Wnt2b, Wnt3, Wnt4, Wnt7b, and Wnt9b, the transmembrane Wnt inhibitors Kremen1 (*Mao et al., 2002*) and APCDD1 (*Shimomura et al., 2010*), and the secreted Wnt antagonist Notum (*Zhang et al., 2015*). We, therefore, chose to further investigate the involvement of the Wnt pathway in GAER signaling by spatially validating the expression of these genes, using the most highly expressed isoforms for mRNA *in situ* hybridization by chain reaction (HCR) probe design.

To characterize the expression of genes encoding Wnt ligands relative to *Shh* expression in the GAER, we used multiplexed ISH by HCR to visualize transcript localization in sections of S26 skate gill arches. We found that while *Wnt7b* was co-expressed with *Shh* in the cells of the GAER (*Figure 3A*), *Wnt2b*, *Wnt3*, *Wnt4*, and *Wnt9b* (*Figure 3B–E*) were all expressed predominantly in the epithelium immediately adjacent to the GAER. We next examined the expression of the Wnt signaling downstream target genes (*Figure 4*) using *Shh* or *Wnt7b* as a marker for the GAER. We found that *Apcdd1*

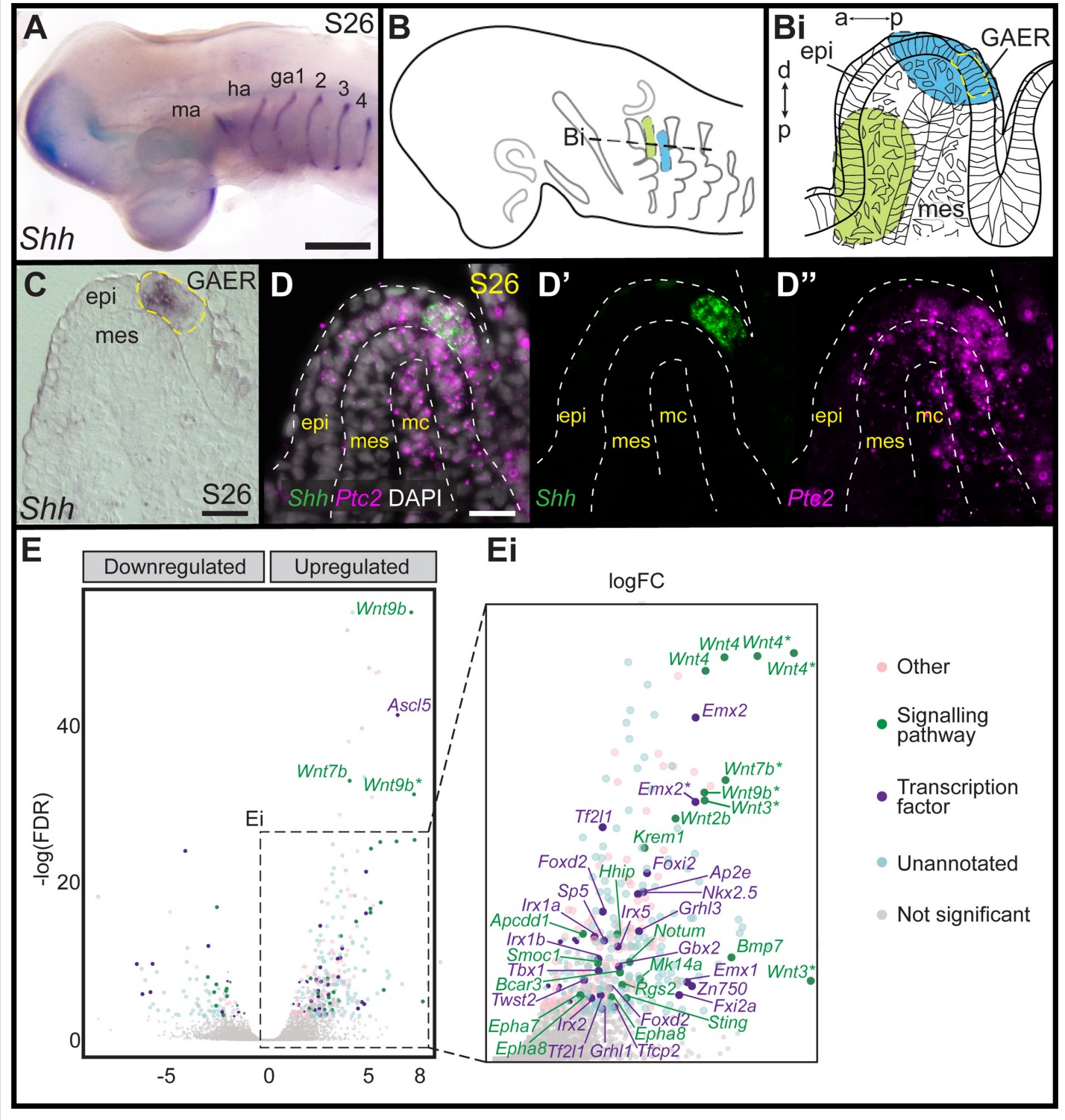

**Figure 2.** Differential gene expression analysis of gill arch epithelial ridge (GAER) and non-GAER gill arch tissues in S26 skate embryos. (**A**) *Sonic hedgehog* (*Shh*) is expressed in the GAER of the hyoid and first four gill arches of skate embryos. (**B**) Schematic of dissection from gill arch one of (**B'**) the non-GAER region (control, green) and the GAER region (blue, with GAER location indicated by yellow dashed outline), as guided by (**C**) mRNA *in situ* hybridization for *Shh* on paraffin section. (**D**) The dissected GAER region includes the *Shh*-expressing GAER, as well as adjacent Shh-responsive (i.e. *Ptc2*+) tissues, as indicated by multiplexed mRNA *in situ* hybridization by chain reaction (HCR) for *Shh* and *Ptc2* on paraffin section. (**E, E'**) Volcano plot illustrating differential gene expression between GAER and non-GAER tissues. Genes with >2 logFC and <0.05 false discovery rate (FDR) are highlighted and assigned functional categories using color coding as per key. Genes with >2 logFC are shown in larger point sizes. Signaling molecules

*Figure 2 continued on next page*

*Figure 2 continued*

and transcription factors with >2 logFC and <0.05 FDR are labeled (* denotes manual annotation of sequence). Differential expression was determined using edgeR with a general linear model and likelihood ratio test, corrected for multiple testing using the Benjamin-Hochberg method to control the FDR. a–p indicates anterior–posterior axis, p–d indicates proximal–distal axis. epi: epithelium, ga1-4: gill arches 1–4, ha: hyoid arch, ma: mandibular arch, mc: mesodermal core, mes: mesenchyme. Scale bars; A: 500 μm C-D: 50 μm.

The online version of this article includes the following figure supplement(s) for figure 2:

**Figure supplement 1.** Genes significantly upregulated in gill arch epithelial ridge (GAER) vs. non-GAER gill arch tissues in S26 skate embryos.

was expressed in the GAER and anterior–distal mesenchyme (*Figure 4A*), *Notum* in the anterior–distal mesenchyme (*Figure 4B*), and *Kremen1* in the anterior (GAER-adjacent) epithelium (*Figure 4C*). Finally, broad transcription of *Axin2* throughout distal gill arch tissues (*Figure 4D*) indicates that Wnt signaling within this territory is occurring through the canonical/β-catenin pathway (*Lustig et al., 2002*). While these findings do not allow us to attribute expression of specific transcriptional readouts to signaling by particular Wnt ligands, our spatial expression data nevertheless indicate that some Wnt signals emanating from the GAER or GAER-adjacent epithelium are transduced preferentially within the anterior arch environment, in a pattern that is largely complimentary to the posterior epithelial and mesenchymal transduction of Shh signals from the GAER.

### *Wnt* and *Shh* expression at the ectoderm–endoderm interface of skate gill arches

The epithelium of vertebrate pharyngeal arches derives from both ectoderm and endoderm, but the specific germ layer origin of the GAER of cartilaginous fishes (endoderm vs. ectoderm) is not known. In the skate, *Shh* is initially expressed broadly in the anterior endodermal epithelium of each pharyngeal pouch, which subsequently gives rise to the posterior endodermal epithelium of each pharyngeal arch (*Gillis and Tidswell, 2017*). To test whether the GAER derives from this initially broad endodermal *Shh* expression domain, we labeled the pharyngeal endoderm of early skate embryos by microinjecting the lipophilic dye CM-DiI into the pharyngeal cavity at stage S18. At this stage, the pharyngeal pouches have not yet fused with the overlying surface ectoderm, allowing for specific CM-DiI labeling of the pharyngeal endodermal epithelium (*Figure 5A and B*). Injected embryos were then grown to S25–S29, by which time the GAER of the hyoid and gill arches is detectable by expression of *Shh* (*Figure 5C*).

Of 29 embryos analyzed, all retained some CM-DiI label within the endodermal lining of the pharyngeal arches. In 8/29 embryos, we recovered CM-DiI labeling up to and including the *Shh*-expressing cells of the GAER in one or more pharyngeal arches (*Figure 5C–C″*), indicating the endodermal origin of these cells. The remaining 21 embryos did not have sufficient labeling of pharyngeal epithelia to inform the origin of the GAER (likely due to label dilution during the long incubation time between injection and analysis). Importantly, however, no CM-DiI labeling was ever observed in epithelial cells immediately anterior to the GAER, suggesting that this epithelium is of ectodermal origin. Together, these observations indicate that the GAER is of endodermal origin and arises at the ectoderm–endoderm interface during pharyngeal arch development and that the *Wnt*-expressing epithelium immediately anterior to the GAER derives from the ectoderm.

### Wnt signaling suppresses GAER Shh signal transduction in the anterior gill arch

To explore the function of Wnt signaling during skate pharyngeal arch development, we inhibited canonical Wnt signaling in skate embryos using IWR1, a small molecule tankyrase inhibitor that antagonizes Wnt signaling by stabilizing the Axin/β-catenin destruction complex (*Lu et al., 2009*; *Kulak et al., 2015*). Briefly, skate embryos were maintained in a bath of 50 μM IWR1 or vehicle-only control (DMSO) in seawater from S25 for 72 hr to assess gene expression changes in response to drug treatment (*Figure 4—figure supplement 1*), or for 6–8 weeks (until S31/32) to assess effects of Wnt signaling inhibition on gill arch skeletal patterning, with replacement of drug- or control-seawater every 48 hr. IWR1-treated embryos showed a marked reduction in the expression of *Axin2* throughout their gill arches (*Figure 4—figure supplement 1A–B*), as well as a reduction in the expression of *Apcdd1* in the GAER and anterior–distal gill arch mesenchyme (*Figure 4—figure supplement 1C–D*)

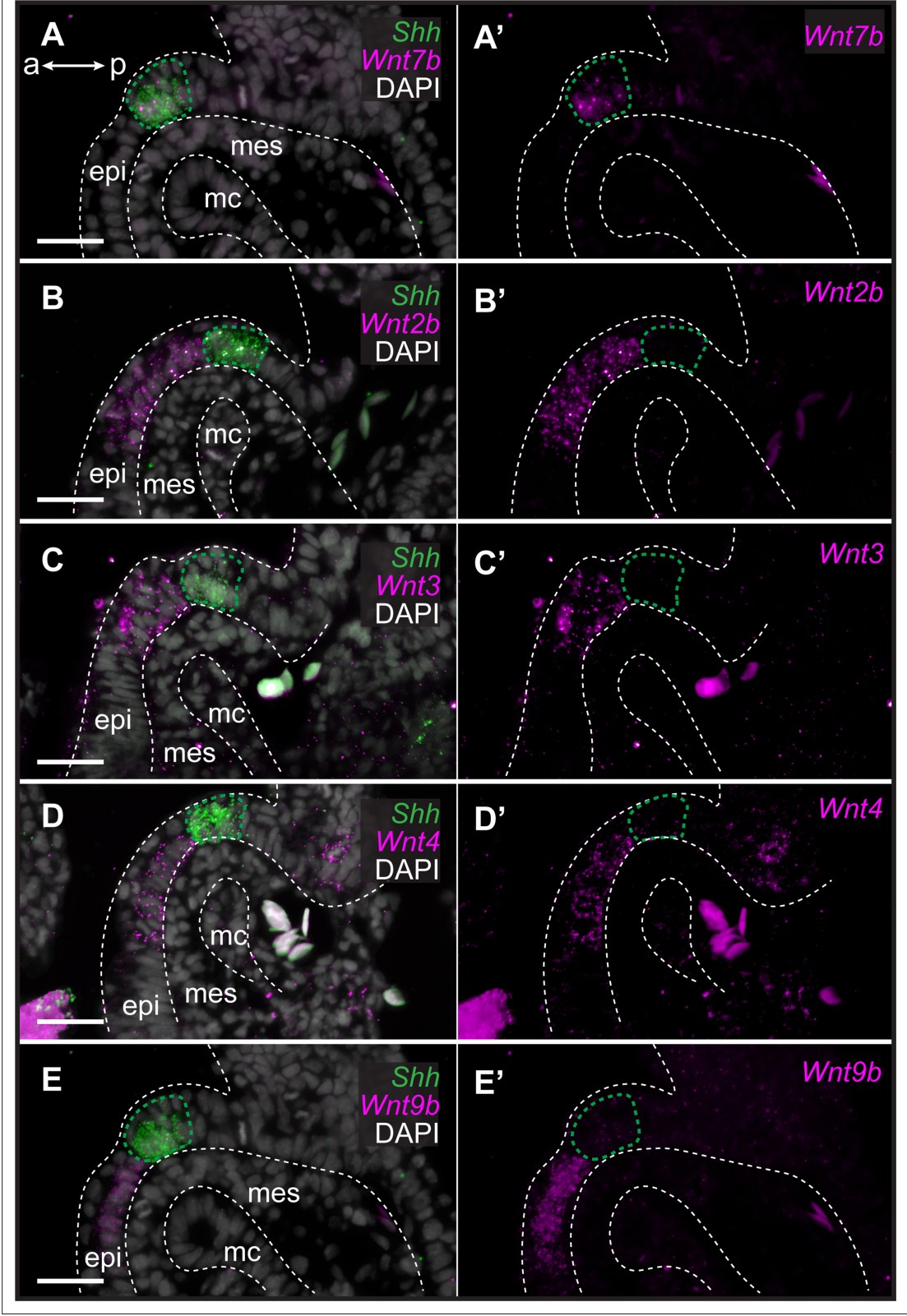

**Figure 3.** Genes encoding Wnt ligands are expressed in and around the gill arch epithelial ridge (GAER) in skate. In S26 skates, (**A, A'**) *Wnt7b* is co-expressed with *Sonic hedgehog* (*Shh*) in the GAER. (**B**) *Wnt2b*, (**C**) *Wnt3*, (**D**) *Wnt4* and (**E**), *Wnt9b* are predominantly expressed in the epithelium immediately adjacent to the GAER. For ease of visualization in **A'–E'**, *Shh* expression is not shown but GAER cells are outlined with a green dashed line. Arch tissues are outlined with white dashed lines. a–p indicates anterior–posterior axis. epi: epithelium, mes: mesenchyme, mc: mesodermal core. Scale bar: 50 μm.

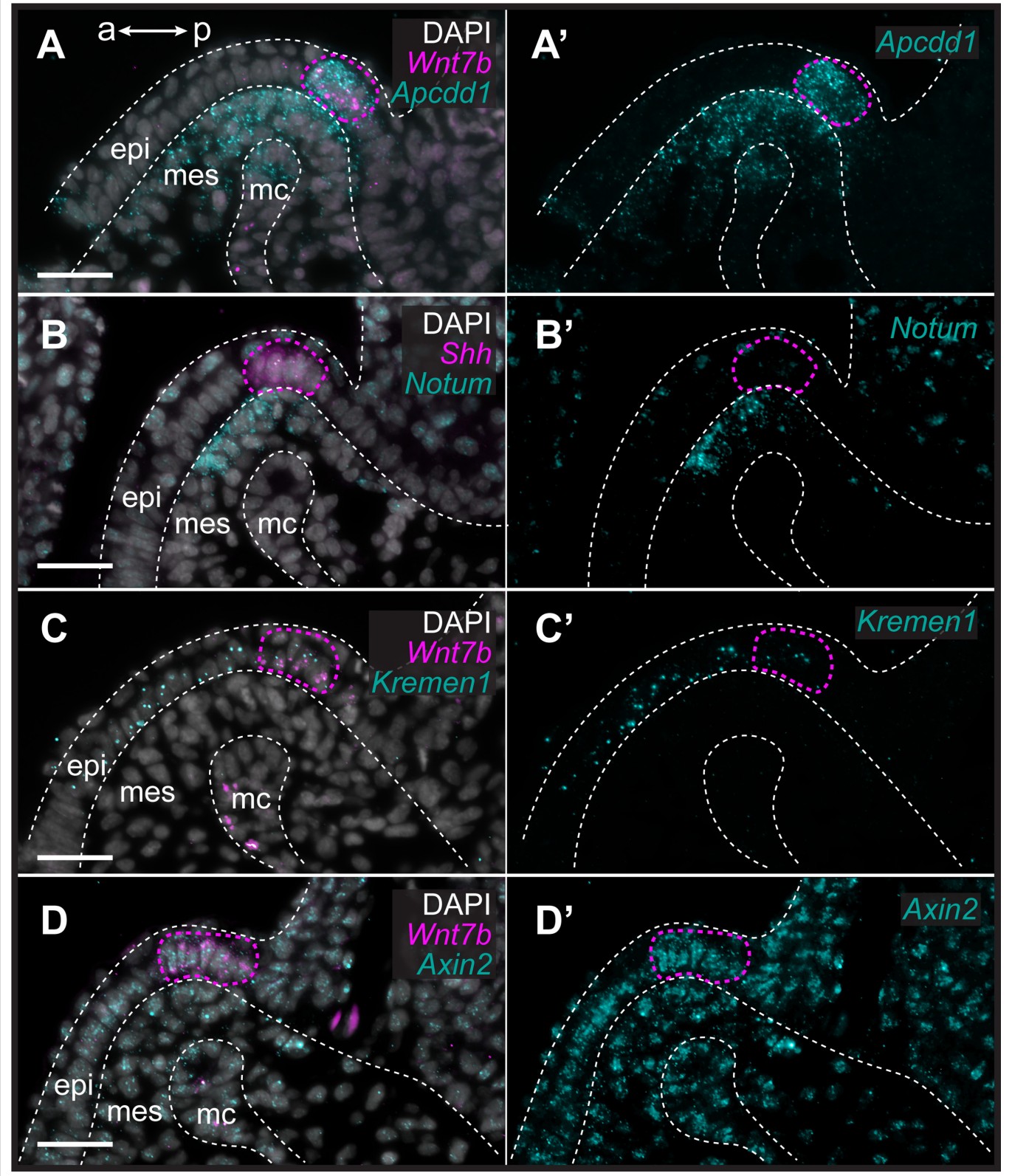

**Figure 4.** Transduction of canonical Wnt signaling in the anterior gill arch territory in skate. In S26 skates, (**A, A'**) *Apcdd1* is expressed in the gill arch epithelial ridge (GAER) and in the anterior-distal mesenchyme underlying the GAER. (**B**) *Notum* is expressed in anterior-distal arch mesenchyme. (**C**) *Kremen1* is expressed in the GAER and in GAER-adjacent epithelium. (**D**) Broad *Axin2* expression in the vicinity of *Wnt*-expressing epithelium indicates signaling through the canonical Wnt signaling pathway. ISH by *in situ* hybridization by chain reaction (HCR) for *Sonic hedgehog* (*Shh*) or *Wnt7b*

*Figure 4 continued on next page*

*Figure 4 continued*

was performed alongside each gene of interest as a marker of the GAER (outlined with a magenta dashed line). a–p indicates anterior–posterior axis. epi: epithelium, mes: mesenchyme, mc: mesodermal core. Scale bar: 50 μm.

The online version of this article includes the following figure supplement(s) for figure 4:

**Figure supplement 1.** Pharmacological inhibition of Wnt signaling in skate leads to downregulation of Wnt pathway genes.

and a complete loss of *Notum* expression in anterior–distal gill arch mesenchyme (*Figure 4—figure supplement 1E–F*). These findings indicate that our drug treatment effectively suppressed canonical Wnt signaling from the GAER and GAER-adjacent epithelium, including those signals transduced asymmetrically in the anterior gill arch environment.

Shh is a pro-chondrogenic signal from the GAER in skate gill arches (*Gillis et al., 2009b*), and this signal is transduced asymmetrically with the posterior gill arch environment (*Gillis and Hall, 2016*, see above *Figure 2D*). To test whether anteriorly transduced Wnt signaling may act to restrict Shh signal transduction to the posterior gill arch, we examined the expression of *Shh* and *Ptc2* in skate embryos 72 hr after the onset of IWR1 treatment (*Figure 6*). While we observed no noticeable change in the expression of *Shh* or in the spatial distribution of *Ptc2* transcripts in gill arch mesenchyme with IWR1 treatment, we did note a qualitative increase in *Ptc2* expression within the normal mesenchymal domain and a clear anterior expansion of *Ptc2* expression within the epithelium adjacent to the GAER (*Figure 6A–D*). To quantify this epithelial expansion, we compared the mean number of *Ptc2+* nuclei

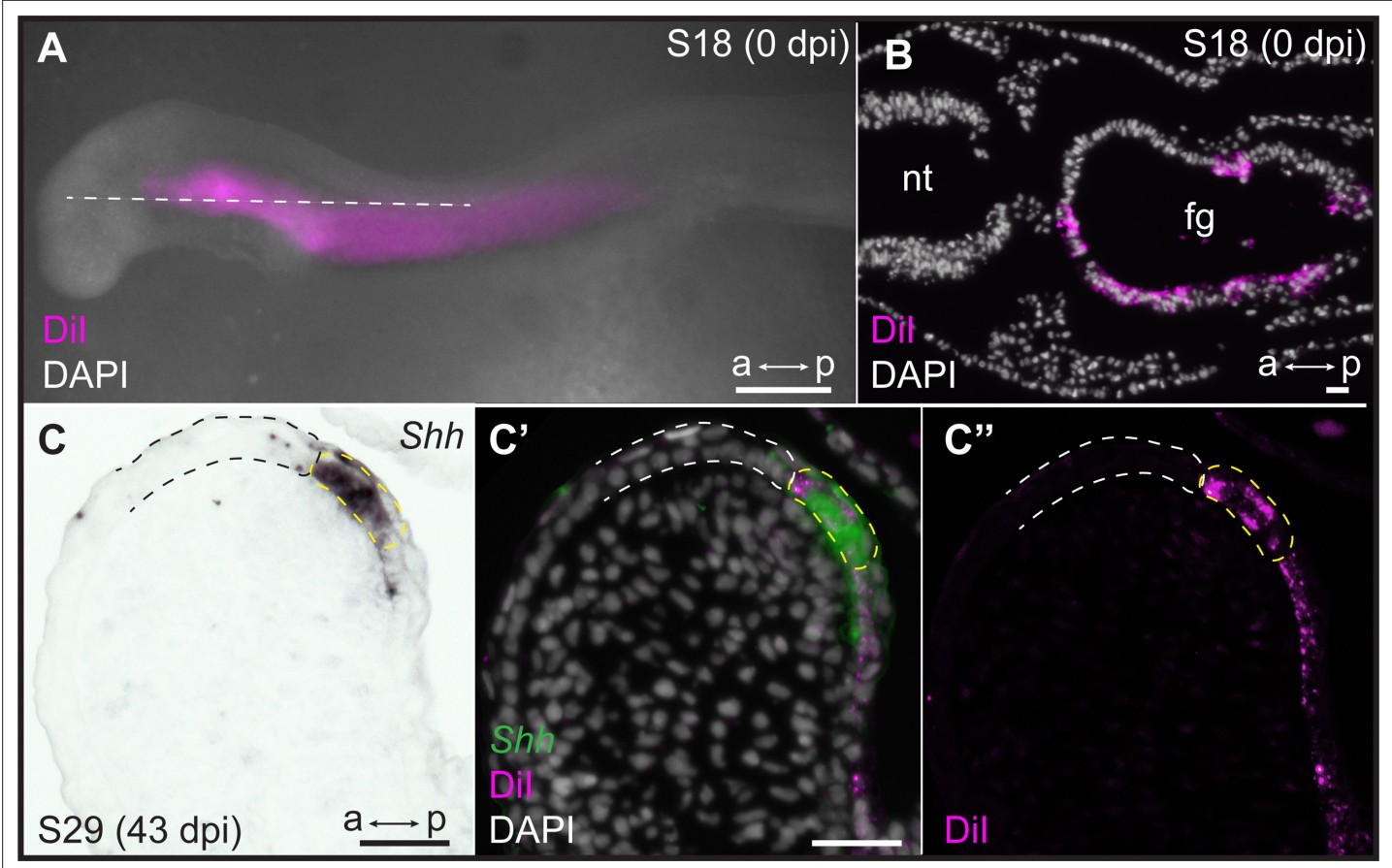

**Figure 5.** Endodermal origin of the skate gill arch epithelial ridge (GAER). (**A**) Microinjection of CM-DiI into the pharyngeal cavity of skate embryos at S18 results in (**B**) specific labeling of the pharyngeal endoderm. (**C**) *Sonic hedgehog (Shh)* is a marker of the GAER (dashed yellow line) in CM-DiI-labeled embryos, and (**C', C"**) co-localization of CM-DiI and *Shh* expression indicates that the cells of the GAER are of endodermal origin. Dashed white outline demarcates GAER-adjacent (ectodermal) epithelium, which is never labeled with CM-DiI. (**C**), (**C'**), and (**C"**) are the same section, imaged sequentially for *Shh* expression and CM-DiI. a–p indicates anterior–posterior axis. dpi: days post-injection, fg: foregut, nt: neural tube. Scale bars; A: 500 μm, B–C: 40 μm.

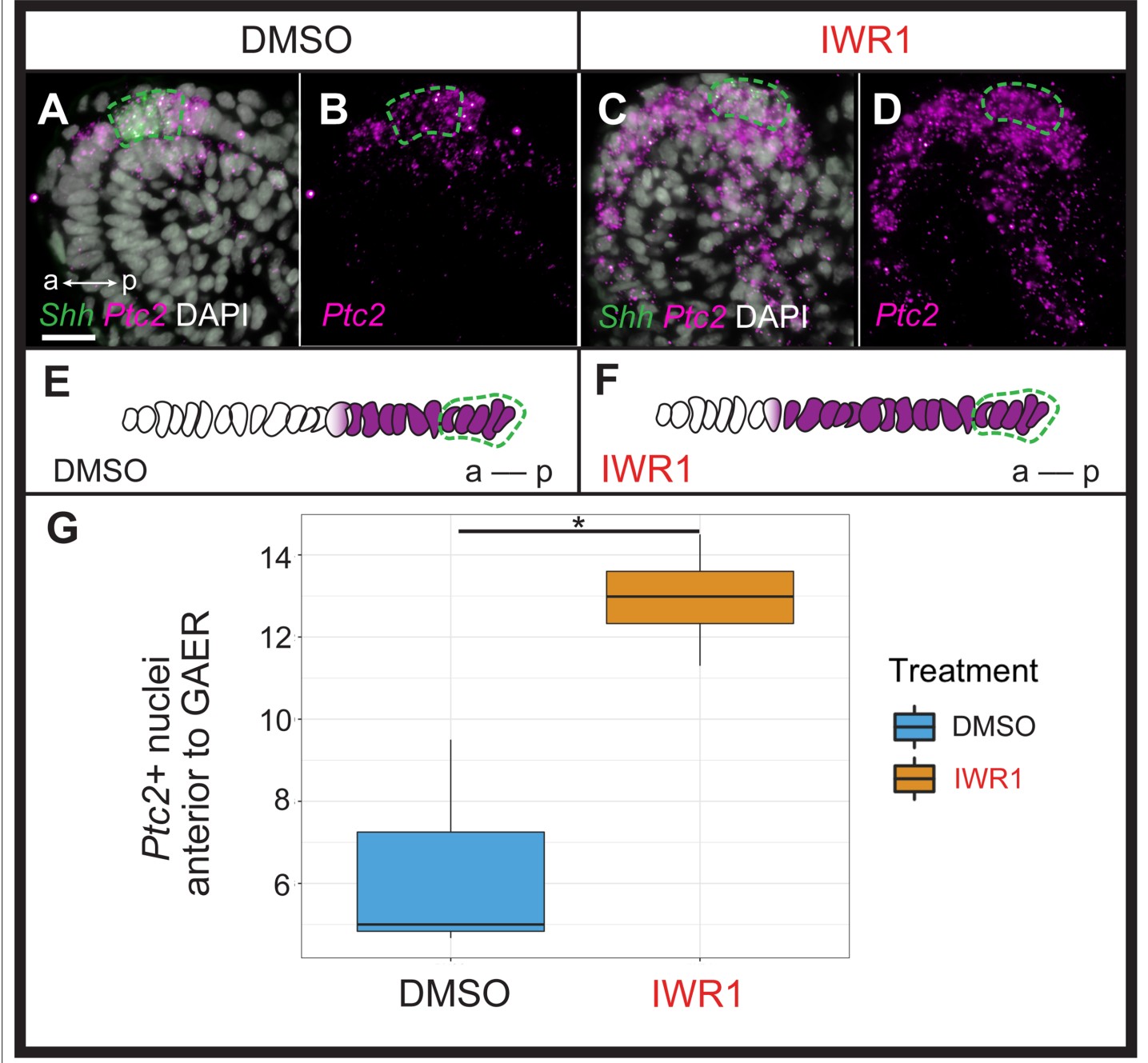

**Figure 6.** Wnt signaling restricts transduction of gill arch epithelial ridge (GAER) Sonic hedgehog (Shh) signaling to the posterior gill arch environment. ISH by *in situ* hybridization by chain reaction (HCR) on sections of skate embryos treated for 72 hr with (**A,B**) DMSO or (**C,D**) IWR1 shows no difference in *Shh* expression but an expansion of *Ptc2* expression within the epithelium anterior to the *Shh*-expressing GAER. (**E**) Schematic illustration of the mean number of *Ptc2*+ nuclei anterior to the GAER in DMSO control (n=3 embryos; mean = 6.3 cells) or (**F**) IWR1 treated (n=4 embryos; mean = 12.94 cells) embryos, using the mean cell count of two or three sections at equivalent positions in the gill arch from each embryo. (**G**) We find a significant increase in *Ptc2* + nuclei in the epithelium anterior to the GAER in IWR1-treated embryos compared to control (p=0.036; t-test, t=–3.87, df = 2.74 - box and whisker plot shows the interquartile range and spread of the data). In **A–F**, *Shh* expression is indicated by a green dashed line. Images in **A–D** were taken using identical exposure settings. a–p indicates anterior–posterior axis. Scale bar: 20 μm.

anterior to the *Shh*-expressing GAER (using DAPI staining and counted manually) across two or three sections at equivalent positions in DMSO control (n=3) or IWR1 (n=4) treated embryos and found a significant anterior expansion of *Ptc2* expression within the epithelium adjacent to the GAER in IWR1 treated embryos (p=0.036; *Figure 6E–G*). This points to a role for GAER-adjacent Wnt signaling in

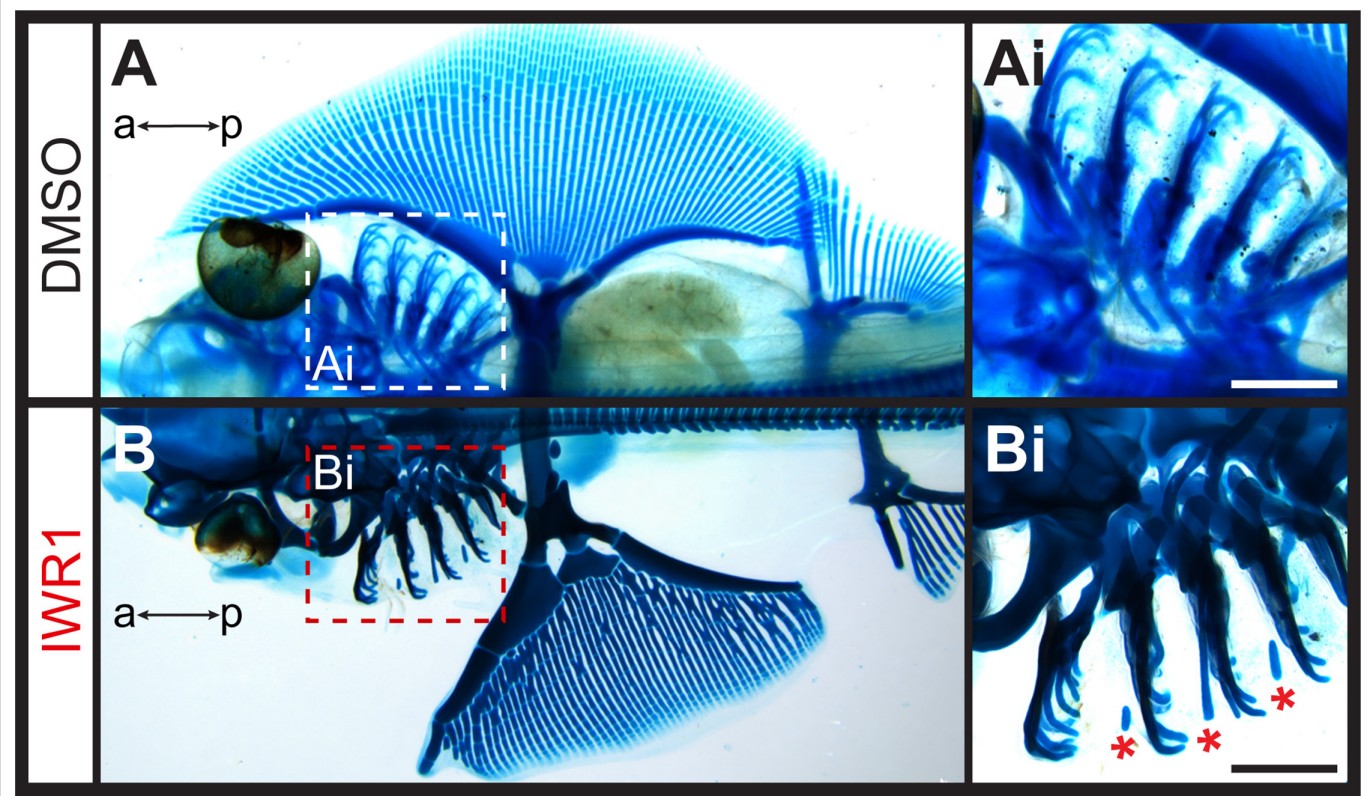

**Figure 7.** Ectopic branchial ray formation upon inhibition of Wnt signaling. (**A**) In control (DMSO) skate embryos, branchial rays (magnified in **Ai**) articulate exclusively with the posterior margin of the gill arches. (**B**) Embryos reared from S25 to S31/32 in the Wnt inhibitor IWR1 possessed ectopic branchial rays (magnified in **Bi**) that were embedded within the connective tissue of the anterior gill arch (ectopic rays present in n=7/8 embryos examined; indicated by a red asterisk). a–p indicates anterior–posterior axis. Scale bars: 500 μm.

restricting the transduction of pro-chondrogenic Shh signaling in the anterior gill arch territory in skate.

## Wnt signaling inhibits branchial ray development in the anterior gill arch environment

Finally, to test for an effect of Wnt inhibition on the gill arch skeleton, we reared skate embryos in DMSO control seawater or IWR1 from S25–S31/32 (*Figure 7*). IWR1-treated embryos had several distinct phenotypic changes, including smaller size, a loss of gill filaments and disrupted fin morphology (the latter consistent with previous findings of *Nakamura et al., 2015*; *Figure 7A–B*). We specifically investigated the impact of IWR1 treatment on gill arch skeletal morphology. The skate hyoid and gill arch skeleton include branchial rays that articulate exclusively with the posterior margin of the gill arch cartilages (*Gillis et al., 2009a* – see also *Figure 1*). These rays also invariably develop on the posterior side of the interbranchial muscle plate – a histologically and molecularly distinct derivative of the paraxial mesodermal core that extends down the middle of each pharyngeal arch (*Daniel, 1934*; *Graham, 2008*; *Hirschberger et al., 2021*). Skate embryos reared in DMSO control seawater showed no gill arch skeletal defects (n=4 embryos; *Figure 7Ai*), while embryos reared in IWR1 seawater possessed conspicuous ectopic branchial rays in the anterior gill arch territory (n=7/8 embryos; *Figure 7Bi*). These ectopic branchial rays occurred along the entire dorsoventral axis of the arch and ranged from short cartilage rods located in the anterior–distal gill arch to nearly full-length branchial rays that extended much of the way toward the gill arch cartilages. In section, branchial rays of skate embryos reared in DMSO control seawater always developed, as per normal, on the posterior side of the mesodermally-derived interbranchial muscle plate of each gill arch: this was evident from histochemical staining and from HCR for the chondrocyte markers *Sox9* and *Col2a1* (*Figure 8A–B*). Conversely, in skate embryos reared in IWR1 seawater, ectopic branchial rays were consistently located

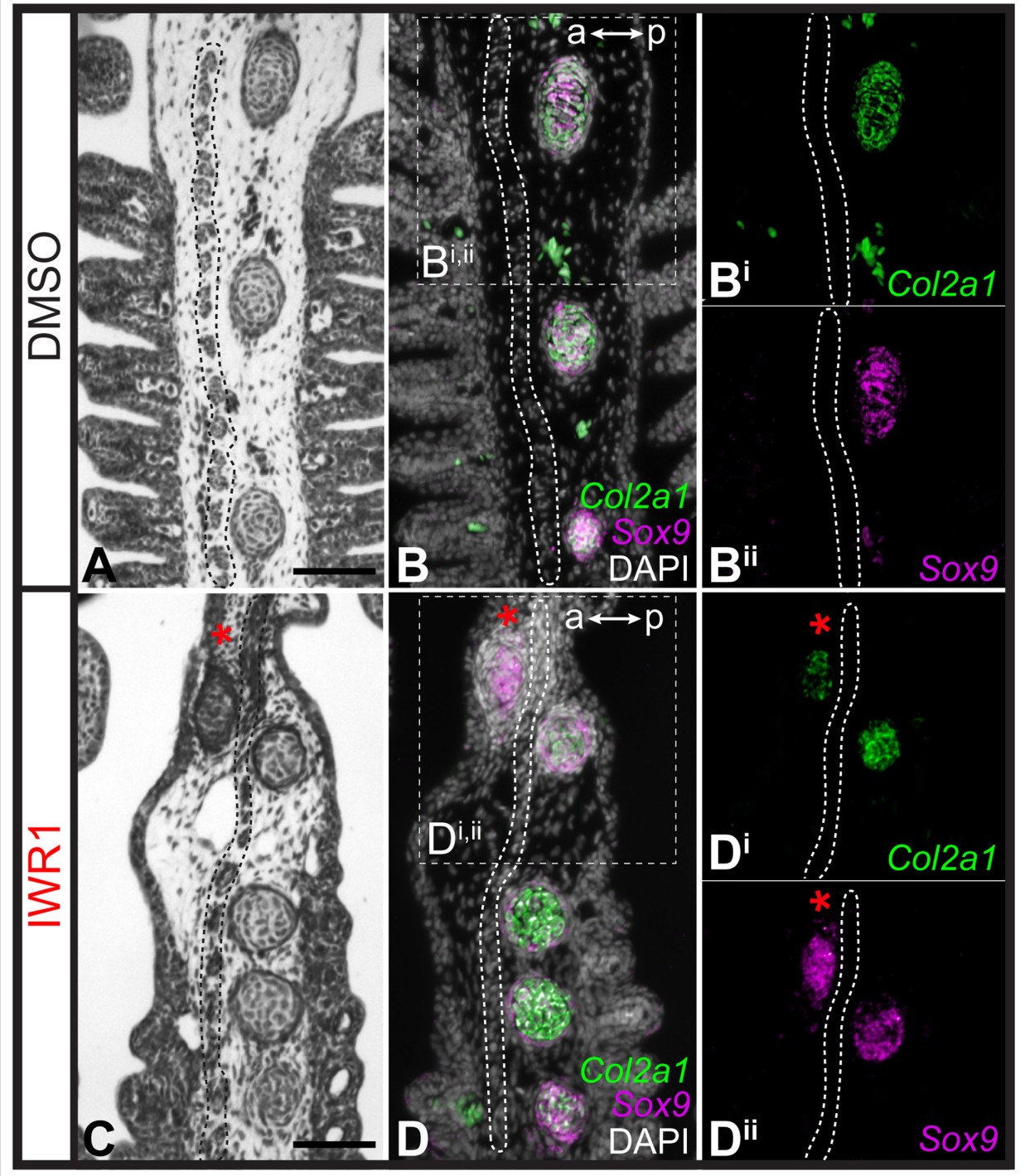

**Figure 8.** Repression of cartilage development in the anterior gill arch by Wnt signaling. (**A**) Section through the gill arch of control (DMSO) S31 skate embryo, showing that branchial rays develop exclusively on the posterior side of the interbranchial muscle plate, as is evident from histochemical staining and (**B, Bi, Bii**) ISH by *in situ* hybridization by chain reaction (HCR) for the chondrocyte markers *Sox9* and *Cola2a1*. (**C**) Section through the gill arch of a S31 embryo reared in the canonical Wnt inhibitor IWR1 reveals that ectopic branchial rays form anterior to the interbranchial muscle plate, as evident from histological analysis and (**D, Di, Dii**) ISH by HCR for *Sox9* and *Col2a1*. Red asterisks indicate ectopic branchial rays, and the interbranchial muscle plate is outlined with a dashed line. a–p indicates anterior–posterior axis. Scale bars: 50 µm.

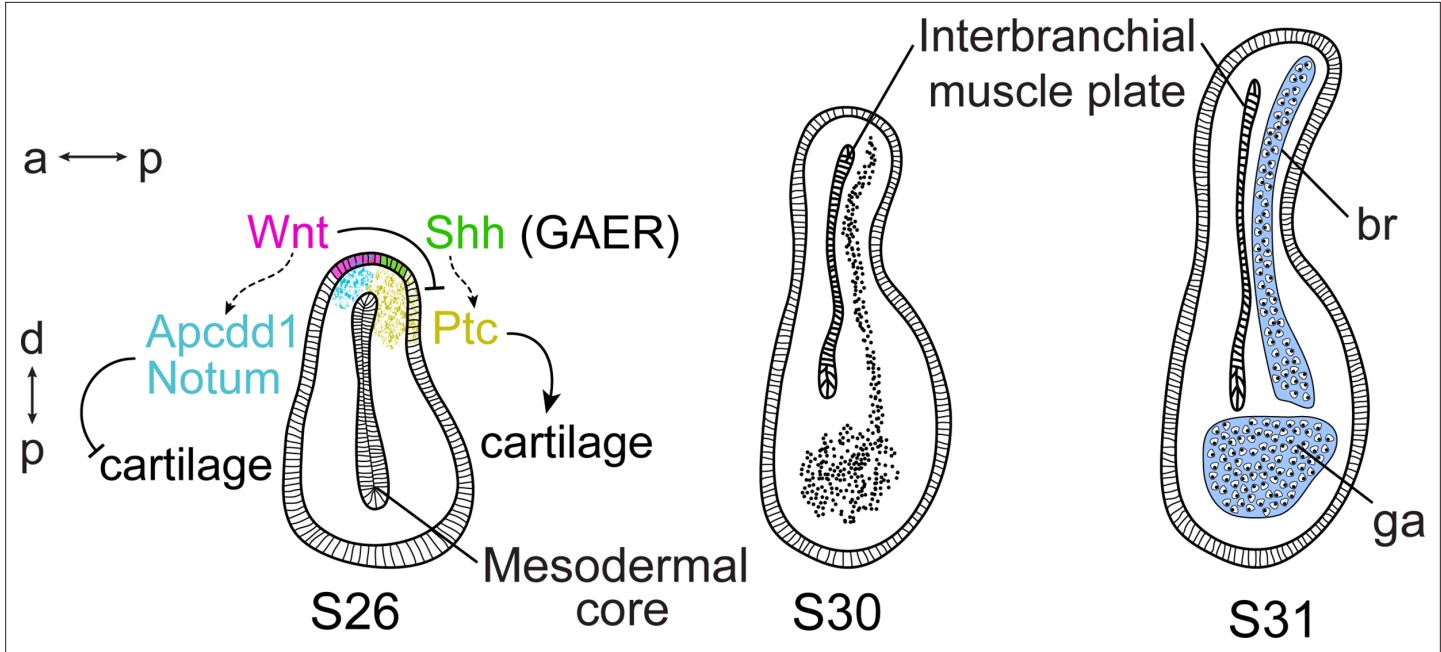

**Figure 9.** Wnt and Sonic hedgehog (Shh) signaling and cell fate determination in skate gill arches. Shh signaling from the endodermal-derived gill arch epithelial ridge (GAER) is transduced within the posterior-distal gill arch (ga) environment, where it promotes the differentiation of cartilaginous branchial rays (br). Wnt signals from ectoderm adjacent to the GAER are transduced within the anterior–distal gill arch environment, where they repress Shh signal transduction and inhibit cartilage formation in the anterior gill arch. a–p indicates the anterior–posterior axis and p–d indicates the proximal–distal axis.

anterior to the interbranchial muscle plate (*Figure 8C–D*). The induction of ectopic branchial rays with IWR1 treatment suggests that Wnt signaling contributes to the maintenance of anteroposterior polarity of the skate gill arch skeleton by repressing chondrogenesis in the anterior gill arch and may do so by restricting transduction of GAER Shh signaling to the posterior gill arch territory.

## Discussion

The molecular mechanisms governing spatial regulation of cell fate decisions are crucial to the establishment and maintenance of anatomical polarity within developing tissues and organs. Here, we show that the GAER signaling centre that forms on the hyoid and gill arches of cartilaginous fishes is of endodermal origin, arising precisely at the boundary of surface ectoderm and pharyngeal endoderm, and that pro-chondrogenic Shh signals from the GAER are transduced preferentially within the posterior gill arch environment. We further show that the ectoderm immediately anterior and adjacent to the GAER is a signaling centre that expresses several Wnt family genes and that Wnt signals from this centre are transduced preferentially in the anterior arch environment. Pharmacological inhibition of canonical Wnt signaling in skate embryos leads to an expansion of Shh signal transduction within the anterior territory of the gill arch, and to the formation of ectopic anterior branchial ray cartilage. These findings suggest that Wnt–Shh signaling antagonism at the pharyngeal ectodermal–endodermal tissue boundary is a key molecular regulator of cell fate determination and anatomical polarity with the developing gill arch skeleton of the skate (*Figure 9*).

Previous analyses have reported the expression of several Wnt genes in developing craniofacial tissues of bony vertebrates (*Summerhurst et al., 2008*), including the expression of *WNT2B* and *WNT9B* in the pharyngeal arch ectoderm of the chick (*Geetha-Loganathan et al., 2009*), *Wnt9b* in the first and second pharyngeal arches in zebrafish (*Jezewski et al., 2008*), *Wnt4a* in zebrafish pharyngeal ectoderm (*Choe et al., 2013*), and Wnt9a and Wnt5b in zebrafish oral epithelium (*Dougherty et al., 2013*; *Rochard et al., 2016*). While β-catenin-independent (i.e. non-canonical) Wnt signaling plays a crucial role in epithelial remodeling during the early formation of pharyngeal endodermal pouches in zebrafish (*Choe et al., 2013*) and in maturation and morphogenesis of the zebrafish oropharyngeal

skeleton (*Rochard et al., 2016*; *Ling et al., 2017*), canonical Wnt signaling has been implicated in patterning of the pharyngeal arch-derived skeleton in multiple taxa. Blocking canonical Wnt signaling causes disrupted facial cartilage formation, including reductions of Meckel's and ceratohyal cartilages in zebrafish (*Alexander et al., 2014*) and loss of ectodermal canonical Wnt signaling in mouse causes loss of facial bones with hypoplasia of all facial prominences (*Reid et al., 2011*). Furthermore, in mouse and human, cleft palate phenotypes may arise as a consequence of mutations in *Wnt3* and *Wnt9b* (*Niemann et al., 2004*; *Juriloff et al., 2006*), and perturbation of Wnt/β-catenin signaling has been implicated in CATSHL syndrome, a human developmental disorder involving craniofacial bone malformation and mispatterning of the pharyngeal arches (*Sun et al., 2020*). Our finding of a role for canonical Wnt signaling in patterning the gill arch skeleton of skate may, therefore, reflect broadly conserved roles for this pathway in pharyngeal skeletal development across jawed vertebrates, though with the skate's possession of branchial rays offering a unique anatomical readout of anterior–posterior polarity defects that arise within the pharyngeal arch-derived skeleton in response to perturbations.

Our findings are also consistent with previously reported complex and context-dependent roles for Wnt signaling in the regulation of cell fate determination events during vertebrate skeletogenesis. In some instances, Wnt signaling functions to promote skeletogenesis: for example, in chick, misexpression of *WNT5A/5B* promotes early chondrogenesis of limb bud mesenchyme *in vitro* and delays the terminal differentiation of growth plate chondrocytes *in vivo* (*Church et al., 2002*), and in mouse, canonical WNT signaling from cranial ectoderm induces specification of osteoblast progenitors of cranial dermal bone within underlying mesenchyme (*Goodnough et al., 2016*). However, in other contexts, Wnt signaling functions to inhibit chondrogenic differentiation or homeostasis. In chick, signaling through WNT3, 4, 7 A, 14 or FZ7 have all been shown to inhibit chondrogenesis *in vitro* or *in vivo* (*Rudnicki and Brown, 1997*), while in mouse conditionally induced haploinsufficiency of the gene encoding the β-catenin degradation complex component APC (i.e. activation of canonical Wnt signaling) results in loss of resting zone chondrocytes and their clonal progeny in developing growth plates (*Hallett et al., 2021*). In the limb bud, genes encoding WNT ligands are expressed in the ectoderm (*Kengaku et al., 1998*; *Parr et al., 1993*; *Geetha-Loganathan et al., 2005*; *Summerhurst et al., 2008*), where they function synergistically with fibroblast growth factor signaling from the apical ectodermal ridge to inhibit chondrogenesis of limb bud mesenchyme and to promote soft connective tissue fates (*ten Berge et al., 2008*). Downstream effectors of Wnt signaling are expressed out-of-phase with *Sox9+* digit progenitors in the mouse limb bud, with Wnt signaling functioning to repress digit chondrogenesis as part of a Turing-like mechanism that underlies periodic patterning of the distal limb endoskeleton (*Raspopovic et al., 2014*). Our findings, therefore, demonstrate a striking parallel between developing skate gill arches and tetrapod limb buds, with Wnt signals emanating from ridge-adjacent epithelia functioning to inhibit chondrogenic differentiation, thereby ensuring differentiation of cartilaginous appendages in a spatially controlled manner.

We have previously shown that Shh signaling from the GAER is required for branchial ray chondrogenesis in skate (*Gillis and Hall, 2016*), and that the application of exogenous Shh protein to the anterior gill arch is sufficient to induce ectopic branchial rays (*Gillis et al., 2009b*). However, it is unclear whether Shh from the GAER is a direct inducer of chondrogenesis in the posterior arch mesenchyme, or rather induces chondrogenesis indirectly, via a secondary signal (i.e. emanating from the Shh-responsive epithelium or mesenchyme). We have observed that inhibition of Wnt signaling in skate embryos has no effect on the expression of *Shh* in the GAER, but rather results in an anterior expansion of Shh signal transduction in the gill arch ectoderm, which in turn, correlates with ectopic chondrogenesis in the anterior gill arch. This observation indicates that the pro-chondrogenic influence of GAER Shh signaling may be indirect, occurring via a secondary Shh-depending epithelial signal whose expression is typically posteriorly restricted by Wnt signaling from GAER-adjacent ectoderm. The Shh and Wnt signaling pathways share multiple downstream components (reviewed in *Ding and Wang, 2017*). For example, components of the canonical Wnt signaling pathway can positively regulate Shh signaling via GSK3β, by phosphorylating the downstream component SUFU and promoting the release of SUFU from Gli (*Takenaka et al., 2007*), and by β-catenin affecting Gli1 transcriptional activity via TCF/LEF (*Maeda et al., 2006*). It has also been found that Gli3 may function as a downstream effector of the Wnt pathway and that Wnt signaling represses Shh activity in the dorsal neural tube through the regulation of *Gli3* expression (*Alvarez-Medina et al., 2008*). Finally,

the Shh signaling pathway may regulate Wnt signaling through Gli1 and Gli2, with these factors positively regulating the expression of the secreted Wnt inhibitor *frizzled-related protein-1* (*sFRP-1*) (*He et al., 2006*). These mechanisms, or others, could account for the apparent cross-regulation between Wnt and Shh signaling that we have observed during the growth and patterning of skate pharyngeal arches.

Finally, it remains to be determined whether antagonistic ectodermal Wnt and endodermal Shh signaling is a derived mechanism of cell fate determination and pharyngeal skeletal patterning in cartilaginous fishes or a conserved feature of the pharyngeal arches of jawed vertebrates. In bony vertebrates, a Shh-expressing posterior endodermal margin (PEM) promotes the proliferative expansion of the second (hyoid) pharyngeal arch (*Wall and Hogan, 1995*; *Richardson et al., 2012*; *Rees and Gillis, 2022*). In bony fishes, the hyoid arch gives rise to an operculum that is supported by a series of dermal bones, and this operculum functions as a protective cover for the gills that arise from the posterior pharyngeal arches, whereas in amniotes, the hyoid arch expands caudally and ultimately fuses with the cardiac eminence to enclose the cervical sinus and close the surface of the neck (*Richardson et al., 2012*). Amniotes lack homologs of the opercular series of bony fishes or the branchial rays of cartilaginous fishes, respectively, but fossil evidence points to the operculate condition of bony fishes as ancestral for jawed vertebrates (*Dearden et al., 2019*). Whether juxtaposed Wnt- and Shh-expressing epithelia influence fate determination (e.g. dermal bone vs. connective tissue) and patterning within the opercular series of bony fishes is not known. Further comparative work between bony and cartilaginous fishes with disparate pharyngeal skeletal architectures and tissue compositions will permit the inference of ancestral functions and interactions of epithelial signals during vertebrate pharyngeal arch skeletal development.

## Materials and methods

### Embryo harvesting

Skate (*L. erinacea*) embryos were obtained from the Marine Resources Center at the Marine Biological Laboratory in Woods Hole, MA, U.S.A., were reared as described in *Gillis et al., 2012* and staged according to *Ballard et al., 1993* and *Maxwell et al., 2008*. Skate embryos were euthanized with an overdose of MS-222 (1 g/L in seawater), and all embryos were fixed in 4% paraformaldehyde (Thermofisher) as per *Gillis et al., 2012*.

### Sectioning and histochemical staining

Embryos were embedded in paraffin and sectioned at 7 µm as described in *O'Neill et al., 2007*. Sectioned embryos were stained with Masson's Trichrome as per *Witten and Hall, 2003* or Mayer's Hematoxylin and Eosin by clearing in two rinses of Histosol (National Diagnostics), rehydration through serial dilutions of ethanol, staining for 15 min in Mayer's Haematoxylin (Sigma), rinsing in running tap water for 20 min, rinsing briefly in 95% ethanol, and staining in 0.1% w/v Eosin Y (Sigma) in 95% ethanol for 2 min. Slides were then washed briefly in 95% ethanol, washed briefly with 100% ethanol, cleared with Histosol, and mounted with permount (Sigma).

### Lineage tracing

Endodermal lineage tracing was performed by microinjection of CellTracker CM-DiI (1,1'-dioctadecyl-3,3,3'3'-tetramethylindocarbocyanine perchlorate; Thermofisher) into the pharyngeal cavity of the stage (S)18 skate embryos with a pulled glass needle. CM-DiI was prepared as per *Gillis et al., 2012*. Labeled embryos were then grown to S25–29, euthanized, fixed, embedded, and sectioned as described above, and CM-DiI localization was assessed relative to the GAER based on mRNA *in situ* hybridization for *Shh* (see below).

### mRNA *in situ* hybridization

Chromogenic mRNA *in situ* hybridization (ISH) was performed on paraffin sections and in whole mount as described in *O'Neill et al., 2007*, with modifications according to *Gillis et al., 2012*. An ISH probe against skate *Shh* (GenBank EF100667) was generated by *in vitro* transcription using standard methods. Following ISH, whole mounts were rinsed, post-fixed in 4% paraformaldehyde, and graded into 75% glycerol for imaging, while slides were rinsed in PBS, post-fixed in 4% paraformaldehyde,

and coverslipped with DAPI-Fluoromount G (SouthernBiotech). Third-generation mRNA ISH by chain reaction (HCR) was performed as per the *Choi et al., 2018* protocol for formalin-fixed, paraffin-embedded sections, with modifications as per *Criswell and Gillis, 2020*. Probes, buffers, and hairpins were purchased from Molecular Instruments (Los Angeles, California, USA). HCR probe set lot numbers are as follows: skate *Shh* (Lot PRA753), *Ptc2* (Lot PRA754), *Wnt2b* (Lot PRE300), *Wnt3* (Lot PRG814), *Wnt4* (Lot PRE301), *Wnt7b* (Lot PRE302), *Wnt9b* (Lot PRE303), *Notum* (Lot PRG817), *Kremen1* (Lot PRG816), *Axin2* (Lot PRG818), *Apcdd1* (Lot PRG815), *Col2a1* (Lot PRB574), and *Sox9* (Lot PRB571). All mRNA ISH experiments were replicated, at minimum, in biological triplicate.

## Pharmacological manipulations

For systemic inhibition of canonical Wnt signaling, experimental and control skate embryos were reared at 15 °C in Petri dishes containing 50 µM IWR1 (Selleck chemicals) in artificial seawater or artificial seawater containing an equivalent volume of vehicle (DMSO) only, respectively. IWR1 has been used previously to inhibit canonical Wnt signaling in skate embryos by *Nakamura et al., 2015* and in shark embryos by *Thiery et al., 2022*. For these experiments, IWR1 was diluted to a working concentration from a 25 mM stock solution in DMSO. Skate embryos were reared in IWR1- or vehicle-containing seawater from S25 until S31/32, with drug or control seawater changes every 48 hr. Once embryos reached the desired stage, they were euthanized, fixed, and processed for whole mount skeletal preparation according to the protocol of *Gillis et al., 2009a* or paraffin histology as described above.

## RNA-Seq analysis of differentially expressed genes in the GAER region

GAER or non-GAER ('control') regions of gill arch 1 were manually dissected from skate embryos at S26 using tungsten needles and flash frozen in lysis buffer using liquid nitrogen. In all cases, GAER and control sample pairs were collected from the same arch within the same embryos, or from opposite arches within the same embryo. RNA was extracted from each sample using the RNAaqueous-Micro Kit. cDNA was synthesized from extracted RNA according to the Smart-seq2 protocol (*Picelli et al., 2014*) and libraries were prepared using the Nextera XT kit (Illumina). Prior to sequencing, barcoded libraries were pooled in equal proportions and checked for quality, insert size, and quantity using Qubit 2.0, and Agilent2100. Sequencing of the 20 libraries generated was conducted by Novogene on an Illumina Hi-Seq-XTen generating 150 bp paired-end reads. The Cambridge Service for Data-Driven Discovery (CSD3) high performance computer was used for cleaning, normalization, quality assessment, assembly, abundance quantification, and annotation, as per *Hirschberger et al., 2021*. Raw reads were cleaned to remove adapter contamination using Trim Galore v0.4.0 (parameters: `--paired` -q 20 –nextera), for quality (Phred score 20) and minimum read length (149 bp) using ea-utils tool fastq-mcf (parameters: -q 20 l 100) (*Aronesty, 2011*). Prior to assembly, all libraries were normalized using Trinity v2.6.6 in silico read normalization (*Haas et al., 2013*) (parameters: --JM 100 G `--max_cov` 75). Trinity v2.6.6 was used to assemble the normalized reads (parameters: `--seqType` fq `--max_memory` 250 G `--CPU` 50 min_contig_length 450) (*Nishimura et al., 2017*; *Langmead et al., 2009*; *Marçais and Kingsford, 2011*). Basic Local Alignment Search Tool (BLAST) was used to compare the assembly to the Uniprot protein database with an e-value cut-off of e-20 to search for sequence similarity (*Altschul et al., 1990*). Transcript abundance was estimated using Salmon, an alignment-free quantification method, via Trinity v2.6.6 utilities (*Haas et al., 2013*; *Patro et al., 2017*). A matrix was built for each of the libraries of gene-level abundance estimates (raw counts) using the Trinity abundance_estimates_to_matrix.pl script. MDS principal component analysis was used to check for batch effects and outliers. Individual embryo origin was a batch effect and so was accordingly accounted for in the additive general linear model. Differentially expressed genes between control and GAER tissue samples were identified using the edgeR_3.28.0 package in R version 3.6.2 (*Robinson et al., 2010*) using a generalized linear model likelihood ratio test (*Chen, 2008*) using a false discovery rate of 5% and no log fold-change (LogFC) cut-off. See supplementary information for assembly statistics, differential expression analysis, and all R scripts used.

Significantly upregulated genes in the S26 GAER tissue (logFC >2, FDR <0.05) were extracted and annotated using blastx sequence similarity searches against the Swissprot/Uniprot database (taxid: *Gallus gallus*). Annotation IDs were inputted into the STRING (v11.5) webserver and tested for predicted protein-protein interactions. In the protein-protein interaction analysis, line thickness

was set to show confidence, and a medium confidence threshold was set. The interaction network was then tested for functional enrichment against a series of databases (Gene Ontology Biological Process, KEGG Pathways, WikiPathways, Pfam Domains, and InterPro Domains) using the default whole genome as the background. Results from the functional enrichment were downloaded and Rstudio (GGplot2) was used to generate dot plots to visualize the most significantly enriched terms.

Differentially upregulated genes with a log-fold change over two were manually screened for genes involved in known signaling pathways and transcription factors to generate a list for candidate gene validation by ISH. Wnt family members were highly upregulated in the GAER region so were validated as candidate markers for this region through third-generation ISH. Transcripts for probes in skate were based on the most highly expressed isoforms.

## Imaging and figure preparation

Images were taken on a Zeiss Axioscope.A1 compound microscope with a Zeiss colibri 7 fluorescence LED light source using a Zeiss Axiocam 305 color or 503 mono camera and ZenPro software or a Leica M165FC stereomicroscope, a Leica DFC7000 T camera and LAS X software. All figures were assembled using Fiji and Adobe creative cloud, with some images flipped or colors inverted for clarity and consistency where needed.

## Acknowledgements

With thanks to Dr Kate Criswell and Dr Christine Hirschberger for advice, and to the University of Cambridge Wellcome PhD. Programme in Developmental Mechanisms. The authors were funded by a Wellcome PhD studentship (214953/Z/18/Z) to JMR, and by a Royal Society University Research Fellowship (UF130182 and URF\R\191007) and Royal Society Research Grant (RG140377) to JAG.

## Additional information

### Funding

| Funder | Grant reference number | Author |
| --- | --- | --- |
| Wellcome Trust | 214953/Z/18/Z | Jenaid M Rees |
| Royal Society | UF130182 | J Andrew Gillis |
| Royal Society | URF\R\191007 | J Andrew Gillis |
| Royal Society | RG140377 | J Andrew Gillis |

The funders had no role in study design, data collection and interpretation, or the decision to submit the work for publication. For the purpose of Open Access, the authors have applied a CC BY public copyright license to any Author Accepted Manuscript version arising from this submission.

### Author contributions

Jenaid M Rees, Conceptualization, Data curation, Formal analysis, Funding acquisition, Investigation, Visualization, Methodology, Writing – original draft; Victoria A Sleight, Data curation, Formal analysis, Supervision, Investigation, Methodology, Writing – review and editing; Stephen J Clark, Methodology; Tetsuya Nakamura, Methodology, Writing – review and editing; J Andrew Gillis, Conceptualization, Formal analysis, Supervision, Funding acquisition, Investigation, Methodology, Project administration, Writing – review and editing

### Author ORCIDs

Jenaid M Rees http://orcid.org/0000-0001-9434-5445
J Andrew Gillis http://orcid.org/0000-0003-2062-3777

### Decision letter and Author response

Decision letter https://doi.org/10.7554/eLife.79964.sa1
Author response https://doi.org/10.7554/eLife.79964.sa2

## Additional files

### Supplementary files

• Transparent reporting form

### Data availability

All data and R scripts for analysis have been deposited on Figshare: 10.6084/m9.figshare.19615779. RNA sequencing data are available at NCBI-SRA under BioProject ID: PRJNA825354. Biosample accessions: SAMN27512544, SAMN27512545, SAMN27512546, SAMN27512547, SAMN27512548, SAMN27512549, SAMN27512550, SAMN27512551, SAMN27512552, SAMN27512553, SAMN27512554, SAMN27512555, SAMN27512556, SAMN27512557, SAMN27512558, SAMN27512559, SAMN27512560, SAMN27512561, SAMN27512562, SAMN27512563. Probes were purchased from Molecular Instruments (Los Angeles, California, USA). This included the following: for skate Shh (Lot PRA753), Ptc2 (Lot PRA754), Wnt2b (Lot PRE300), Wnt3 (Lot PRG814), Wnt4 (Lot PRE301), Wnt7b (Lot PRE302), Wnt9b (Lot PRE303), Notum (Lot PRG817), Kremen1 (Lot PRG816), Axin2 (Lot PRG818), Apcdd1 (Lot PRG815), Col2a1 (Lot PRB574) and Sox9 (Lot PRB571).

The following datasets were generated:

| Author(s) | Year | Dataset title | Dataset URL | Database and Identifier |
|---|---|---|---|---|
| Gillis JA | 2023 | Ectodermal Wnt signaling, cell fate determination and polarity of the skate gill arch skeleton | https://www.ncbi.nlm.nih.gov/bioproject?term=PRJNA825354 | NCBI BioProject, PRJNA825354 |
| Rees JM | 2023 | Scripts and data on Wnt signalling, cell fate determination and anteroposterior polarity of the skate gill arch skeleton | https://doi.org/10.6084/m9.figshare.19615779.v2 | figshare, 10.6084/m9.figshare.19615779.v2 |

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
