## [Editor Report]

Building on previous work dissecting the polarisation of a gill arch epithelial ridge (GAER) of a cartilaginous fish (skate), this paper uses RNA–Sequencing, multiplexed HCR, cell fate mapping, and pharmacological manipulation to uncover the contribution of the Wnt signalling pathway to the patterning of the gill arch, highlighting the role tissues boundaries in signalling and patterning embryonic tissues.

---

## [Decision Letter]

**Decision letter after peer review:**

Thank you for submitting your article "Wnt signalling, cell fate determination and anteroposterior polarity of the skate gill arch skeleton" for consideration by *eLife*. Your article has been reviewed by 3 peer reviewers, including Megan G Davey as the Reviewing Editor and Reviewer #1, and the evaluation has been overseen by Didier Stainier as the Senior Editor. The following individuals involved in the review of your submission have agreed to reveal their identity: Ruth Williams (Reviewer #2); Shawn A. Hallett (Reviewer #3).

Essential revisions:

1) The paper requires significant re–writing in order to improve its accessibility, including combining figures and reducing the number of figures; shortening and improving the introduction; improving figure annotation (scale bars, labelling, etc) and improving summary figures; addition of key missing references i.e. Wnt signaling in the developing zebrafish craniofacial complex (10.1038/s41598–021–85415–y, 10.1016/j.ydbio.2016.11.016, 10.1242/dev.137000); justification of the selection of Ptch2 and Dusp6 as downstream markers of Shh and Fgf8; improved recording of statistical methods and acknowledgment of how tissues were correctly dissected. Please reference the extensive suggestions from all the reviewers.

2) While I do not think that additional wet experimental work beyond that supporting the current data is required, there is wet work that should be undertaken to support current data and which may benefit the authors and allow them to reduce the complexity of the figures. Including–

– Expression data validating the molecular profile of "mesoderm" cells should be included.

– Histologic evidence that the tissues they are isolating are truly what they appear to be. For example, following manual dissection of similarly staged samples, remaining tissue should be stained with markers for the GAER versus non–GAER.

– Multiplexed HCR RNA in situ hybridisation, in order to fully explore co–localisation of genes expressed in the GAER and non–GAER tissues i.e. SHH and FGF8.

– In addition, to support the transcriptomic approach, it would be an improvement to see the expression of other components of the FGF and WNT signalling pathways, such as Wnt2b, Wnt3, Wnt4, and Wnt9b, FGF receptors, 1/2/3.

– Multiplexed HCR RNA in situ hybridisation to underpin findings from the transcriptomics, for example, gene expression of tissue–specific transcription factors would also significantly lift the paper, but it is acknowledged that tissues, time, and expense may prevent this.

3) If available for the little skate, the authors should perform KEGG analysis and gene ontology of data from the transcriptomic data set.

*Reviewer #1 (Recommendations for the authors):*

I like your paper but it was overly complex. Not many people understand the branchial arch formation and even less the nuances of gill arch anatomy and the difference between cartilaginous and non–cartilaginous fishes. I feel your Intro is really bogged down in the complexities of cartilaginous and non–cartilaginous fishes and gill arches – whereas your premise is really pretty simple. "These signalling pathways work together to make a conserved patterned structure". Basically, spell out what the interesting point of the paper is, and frame the question. And why skate? At present, it is overly complex.

To improve the paper I think an intro with some back–to–basics would help considerably. If you need the reader to understand the differences between the ancestral condition, cartilaginous Vs bony fish gill arches, then perhaps consider an intro drawing showing this, highlighting the pattern you are interested in. I would also add an intro figure outlining your hypothesis – perhaps something like Figure 5Bi or Fig10, showing the signalling centres, rather than the factors they express. Do you actually need a drawn figure which compares chicken and skate tissues and signalling centres?

Also, I think there are some incorrect sentences in places, which make the paper confusing, I assume they are just editing mistakes but do make the paper tricky to read e.g.

"All jawed vertebrates belong to one of two lineages: cartilaginous fishes (sharks, skates, rays, and holocephalans) or bony fishes (ray– and lobe–fined fishes…)" But not all jawed vertebrates are fish (?)

250 The epithelium of vertebrate pharyngeal arches derives from both ectoderm and endoderm, and the specific tissue origin of the GAER of cartilaginous fishes is not known."

But isn't a cartilaginous fish a vertebrate? And the GAER is the epithelium of a branchial arch–derived structure– so should be from ectoderm and endoderm? I appreciate this might not have been shown but your sentence is confusing. Should it be "The epithelium of vertebrate pharyngeal arches derives from both ectoderm and endoderm? While this suggests the specific tissue origin of the GAER of cartilaginous fishes is probably ectoderm and endoderm this has not been shown."?

Figure1 is beautiful – but really needs annotation. In A', I presume we are looking at the gill arch? In B' Would be better to annotate which are the branchial rays, the epibranchial cartilage, and the ceratobranchial cartilages. And how the branchial arches are polarised. Obvious to you, I'm sure. But not so much to the rest of us.

Figure3 – label the endoderm for clarity.

I think the fate mapping data and gene expression data for the chick PEM would be better in the main text to better support your conclusion that "These findings demonstrate that the skate GAER and chick PEM share embryonic tissue origins and expression patterns of signalling molecules through development and are therefore homologous and an ancestral feature of the pharyngeal arche(s) of jawed vertebrates.", although it does make me wonder why, if the structures are homologous that you didn't use the chick more…or at least test your Wnt hypothesis in the chick with IWR1 treatment.

I would recommend publication but I think you should re–write in a style that is easier to understand and perhaps reorganise your data to show chicken and ray side by side.

*Reviewer #2 (Recommendations for the authors):*

Very nice work. An interesting read. I just have a few specific comments about the presentation of the figures below. If feasible, it would be very nice to see HCR used to its potential for multiplexing, this would condense the figures and therefore improve flow, and indeed the overall length – which at 10 figures is off–putting from the outset.

I've also commented that RNA–seq on treated embryos, more specific Wnt modulation, and equivalent experiments in chicks would be nice. I understand these are expensive, and in the case of the treatment regime, time–consuming, so perhaps something for a 'research advance' on this work.

And just a general comment/thought, going forward, it would be interesting to find out if there is a tissue–specific enhancer for Shh in the GAER, similar to that in the limb bud and pectoral fin. This would provide a valuable tool to drive tissue–specific gene manipulation and also isolate this specific cell population.

Figure 1: Please help the reader by orientating what is shown in B. Where is this in A? Please provide a scale bar here too. In panel C, the legend is not clear, do the asterisks label the hyoid and gill arches, or just gill arches – but 5 asterisks are present and only 4–gill arches are mentioned in the legend? Maybe label the hyoid differently to be clear. Also, is Shh expressed in a lower jaw structure? In panel D, please annotate A/P and D/V orientation. Ideally, an additional gill arch marker should be shown to demonstrate the posterior regionalisation of Shh. HCR would also give a better resolution.

Figure 2: Please label the A–P axis on the figures, as mentioned in the legend. This is actually lacking throughout the figures and would be very helpful to include.

Figure 3/4: It would look much nicer if Shh and Fgf8 co–expression at S22 and S29 were also shown by HCR in figure 4, making one concise in situ figure instead of two. Also, co–expression of all 4 markers Shh, Fgf8, Ptc2, and Dusp6 could be shown in one HCR experiment. On the HCR images, please show the outline of the gill arches with a white dotted line, as figure 2C.

Figure 2 supplement 1: I think it would read better if the chick lineage tracing section came immediately after the skate lineage tracing. In panel A, it would be useful to indicate the location of a whole embryo in this section is taken at. Ideally, Shh HCR should be shown on the same embryo with CCFSE staining, or is this not possible of CCFSE–treated embryos? I think this figure could be merged with the main figure 2. For example, panel E can be removed and only E' is needed. E", Ei, and Ei' are not necessary, this labelling is not logical and they are not referred to in the legend. In main figure 2, panel C is not necessary, so I think these can be easily combined.

It seems only 24–31% of embryos injected with DiI had the same labelling in the GAER, is this proportion expected with DiI labelling?

Figure 3 supplement 1: It seems this is more a supplemental figure for figure 2 since it pertains more to the lineage tracing. I think the U panel of HCRs could join a new combined figure of Figure 2 and Figure 2 supplement 1.

In figure 3 supplement 1, panels B, G, L, and Q need to have more structures labelled, as described in the legend. These in situs would be so much clearer by multiplexed HCR.

Figure 5: Did you do gene ontology analysis for the upregulated genes? What were the general pathways that were potentially perturbed, do you see Wnt signalling for example? Also, can you tie this in better with the in situs shown in previous figures, eg do you see Shh, Dusp6, Ptc2, and Fgf8 upregulated in the GAER?

Figure 6: Did you observe any general phenotype on the IWR1 and DMSO treated embryos? I see now in figure 8, that the embryos are very deformed. Perhaps you can comment on the general phenotype and broad range effects of the IWR1 treatment.

Please show the Shh expression in green, not just the green dotted line. If this clouds the magenta signal of the Wnt molecules please show separate colours in different panels.

Did you use HCR to validate any of the transcription factors?

It would be most useful to conduct RNA–seq on GAER dissections from IWR1 treated embryos. But not essential for revision.

Did you attempt a more specific disruption of Wnt signalling by way of CRISPR mutation? Also, did you examine the effects of disrupted Wnt signalling on the chick PEM?

Figure 7: Please label A–P. How many sections were examined from each embryo? How were Ptc+ cells counted, how did you define a cell, and did you use a membrane stain or Dapi? In the figure legend, please quote the t–test result correctly, eg what are the degrees of freedom and t statistics.

Figure 9: This is a very interesting specific phenotype in an otherwise broadly affected 'mutant'. Were *Sox9* and Col2a1 expression examined at earlier stages? Please quote the embryo stage in the legend.

Figure 10: This is a very nice schematic summary of the key findings presented here, but please annotate further eg A/P, D/V, label GAER, and surrounding tissues. It would be very useful to include this schematic in figure 1 to give the reader better orientation of the tissues from the outset, then use the same schematic in figure 10 to show your findings. A similar schematic of the chick PEM would also help to demonstrate the similarities between these tissues.

*Reviewer #3 (Recommendations for the authors):*

– Although it may be useful to orient the reader, Figure 1 is largely extraneous and unnecessary as there have been several studies cited that provide the same expression data.

– Figure 3. Although histologically one can discern between the epidermal–mesodermal boundary, some expression data validating the molecular profile of "mesoderm" cells should be included.

– Figure 4. Additional justification in the text should be included to inform the reader of the selection of Ptch2 and Dusp6 as downstream markers of Shh and Fgf8, respectively.

– Figure 4. The authors should assess expression domains of the FGF receptors, 1/2/3, as they have been shown to be critically important in chondrogenesis in other models and to discern the target of Fgf8.

– Figure 4. Additional literature review highlighting the potential differential roles of other FGF ligands in brachial arch formation should be included in the text and potentially assessed on an mRNA level histologically.

– Figure 5. How do the authors differentiate cell types during their isolation method in preparation for transcriptomic analysis? To be sure of the accuracy of their methods, the authors should provide histologic evidence that the tissues they are isolating are truly what they appear to be. For example, following manual dissection of similarly staged samples, remaining tissue should be stained with markers for the GAER versus non–GAER.

– Figure 5. If available for the little skate, the authors should perform KEGG analysis on transcriptomes of their data set.

– Figure 6. If mRNA transcripts of Wnt2b, Wnt3, Wnt4, and Wnt9b are truly restricted to non–GAER regions, the authors should provide more conclusive evidence, for example by co–staining these sections with Shh. The current diagram seems slightly arbitrary, as the GAER location may vary across tissue sections.

– Figure 9. It would be interesting to further elucidate the state of the *Sox9*/Col2a1–expressing. A more extensive analysis of gene expression profiles would be important to further explain the ectopic brachial arch phenotype (e.g. Pthrp, Ihh, Col10).

– Discussion. Several papers have explored the role of Wnt signaling in the developing zebrafish craniofacial complex (10.1038/s41598-021-85415-y, 10.1016/j.ydbio.2016.11.016, 10.1242/dev.137000). These should be considered and included.

[Editors' note: further revisions were suggested prior to acceptance, as described below.]

Thank you for resubmitting your study entitled "Ectodermal Wnt signalling, cell fate determination and polarity of the skate gill arch skeleton" for further consideration by *eLife*. Your revised article has been evaluated by Didier Stainier (Senior Editor) and a Reviewing Editor.

The reviewers are all in agreement that the manuscript has been improved and merits acceptance based on scientific merit but there are some remaining issues that need to be addressed. These are primarily to make the paper accessible. My feeling is that Figure 1 is overly complex and, as it is primarily an introduction figure, could be improved and/or reduced and combined with Figure 2.

Figure1 problems–

The panels are not in the same orientation, which makes it confusing to the reader.

– Figure 1A is a very stunning picture– but it does not explain the gill arch anatomy in relation to the brachial ray anatomy. It would be better to have a drawn picture I think.

– Figure 1A'. Not mentioned in the figure legend.

– Figure 1B– this is the Anterior–posterior polarity the paper is focused on, but to the uninitiated, what the polarity is in the figure, is unclear and isn't easy to associate with the text. There is no A–P marked in this panel. Is this figure really the same orientation as A and A'? I don't understand the orientation of these samples at all, doesn't the axis of the gill arch run medial–lateral?

– Figure1C. Again a different orientation. what is lateral and what is medial?

– Figure 1D – the asterisk is not defined in the figure legend (it is in the body of the text). But if the asterisk marks the SHH domain in the lateral part of the gill arch, what is purple in the medial part of the gill arch? Why is the GAER not annotated here? Why is M–L not annotated? What about the HA?

Figure F – Is this a summary of the gill arch? This is not stated in the figure or the text. 'distal' is added as a term. Does distal=lateral? The text for F is incomplete "…where is promotes differentiation of cartilaginous branchial'…branchial what?

Figure 2 is a much better figure. I think that Figures 1 and 2 should be combined, keeping Figure 1B, C, E–E' and all of figure2. For example, Figure 2C. is a (better) version of 1D. My main confusion now is that in Figure 2, M–L seems to be reversed from Figure 1. Is this correct?

These comments also relate to Fig7 and 9 which either do not annotate A–P or have M–L in a different orientation.

Reviewer 2 also comments that they are unsure about Figure 5A– the DiI labelling. I think this is Figure 7? But as the figures are not numbered we cannot be sure. please check and revise if incorrect.

Other confusing points in the paper– is the gill arch the same as the pharyngeal arch? These terms are used interchangeably in the paper, for example in the first two sentences. But I am not sure if they are equivalent.

Other confusing points in the paper- is the gill arch the same as the pharyngeal arch? These terms are used interchangeably in the paper, for example in the first two sentences. But I am not sure if they are equivalent.

The other very positive comments are below-

*Reviewer #1 (Recommendations for the authors):*

The manuscript is much improved for removing the chick data. The figures flow better and the annotation is also much improved (although they were not numbered in the document provided here, and I think the Dil labelling figure is not in the correct position).

The data provides a good characterization of gene expression in the GAER region and non-GAER tissue which, at least for Wnt factors has been beautifully validated by multiplexed HCR.

Lineage tracing provides novel insight into the embryonic origin of these tissues.

Overall, I think the manuscript is significantly improved and the data offers a useful resource for other researchers as well as providing a platform for further, more extensive analysis of signaling events from the GAER underlying gill development, which I look forward to reading about!

*Reviewer #3 (Recommendations for the authors):*

The authors have significantly improved the clarity and quality of the writing and data, respectively, included herein, by highlighting key references and further expanding on their transcriptomic analyses in the introduction and discussion sections. Importantly, by placing an emphasis on Wnt and Hedgehog signaling, the authors have eloquently demonstrated a critical role of these pathways in establishing gill arch skeletal polarity in skates. I have no further suggestions at this time.

---

## [Author Response]

Essential revisions:1) The paper requires significant re–writing in order to improve its accessibility, including combining figures and reducing the number of figures; shortening and improving the introduction; improving figure annotation (scale bars, labelling, etc) and improving summary figures; addition of key missing references i.e. Wnt signaling in the developing zebrafish craniofacial complex (10.1038/s41598–021–85415–y, 10.1016/j.ydbio.2016.11.016, 10.1242/dev.137000); justification of the selection of Ptch2 and Dusp6 as downstream markers of Shh and Fgf8; improved recording of statistical methods and acknowledgment of how tissues were correctly dissected. Please reference the extensive suggestions from all the reviewers.

As discussed above, we have significantly revised and streamlined the text, figures, and figure annotations throughout, in response to the detailed feedback of all three reviewers. All three reviewers made suggestions of different ways in which the manuscript could be improved. We’ve taken guidance from all three reviews, but in some instances, changes requested by one reviewer were at odds with suggestions from another (cutting vs combining and expanding figures, sections of text to expand/reduce, etc). Rather than going through a point-by-point list of the many individual edits that we’ve made (and given the scale of the revision), we hope that you will consider this revised submission in total and will recognised the many places in which reviewers’ comments greatly helped to improve the clarity of the text, figures and message.

In our revised discussion, we have included additional references (Lines 403-414), including some of the ones recommended by Shawn Hallett in his review. There are hundreds of papers dealing with Wnt signalling and skeletal development, and we can’t really review them all in this paper. We have opted to focus largely on discussing the role of Wnt signalling in cell fate determination (rather than the role of non-canonical Wnt signalling in morphogenesis and convergent extension). We have been selective about which new references to include to keep with the focus of our paper. But we have also taken your suggestion and included additional new references relating to the *Sox9*-Wnt-BMP Turing-like mechanism that underlies amniote digit patterning, and we discuss parallels with the anti-chondrogenic role for Wnt signalling in skate branchial rays that we report here (Lines 449-456).

We have added a reference to Pearse et al. (2001), to support the use of *Ptc2* as a transcriptional readout of Shh signalling (Lines 71-74). We no longer discuss Fgf8 signalling in this manuscript, and so have not added additional references relating to *Dusp6*.

In the legend to Figure 6 (formerly Figure 7), we have improved reporting of our statistical methods. We have specified that we counted *Ptc2*+ nuclei (based on DAPI staining) and compared the means of 2-3 sections from equivalent positions in the arches of DMSO control (n=3 embryos) and IWR1-treated (n=4 embryos) animals. We also now include the t value and the degrees of freedom in the legend.

2) While I do not think that additional wet experimental work beyond that supporting the current data is required, there is wet work that should be undertaken to support current data and which may benefit the authors and allow them to reduce the complexity of the figures. Including–– Expression data validating the molecular profile of "mesoderm" cells should be included.

We presume that the mesoderm referred to here is the interbranchial muscle plate, which derives from the paraxial mesodermal core that runs down the center of the pharyngeal arch. This mesoderm-derived muscular sheet is readily identifiable histologically and also exhibits distinct gene expression features that have been reported elsewhere (most recently, in skate, by us in Hirschberger *et al.,* 2021, Mol. Biol. Evol. 38: 4187). In our figures, these muscle fibres are clearly recognisable with regular histochemical staining (in revised Figure 8), and we don’t really think that we could easily include additional panels to illustrate gene expression features of this muscle sheet into that plate. But we have added new citations of a classic anatomical text, a review article on pharyngeal arch tissue structure and our study on gene expression in pharyngeal arch tissues in skate to further support and clarify the nature and location of this tissue within the arch (Lines 347-350).

– Histologic evidence that the tissues they are isolating are truly what they appear to be. For example, following manual dissection of similarly staged samples, remaining tissue should be stained with markers for the GAER versus non–GAER.

Our manual microdissection of embryonic skate gill arches was guided by arch morphology, and by our extensive experience working with this tissue. We first used sharpened tungsten needles to dissect the posterodistal GAER region (which appears as a morphologically distinct pseudostratified epithelial ridge along the leading edge of the expanding arch), and then removed a region of non-GAER tissue from the anterior face of the same arch. By the time these dissections were finished, the arch had been heavily dissected, and so we did not save or fix the remaining tissue. But to confirm that we did effectively capture the GAER region, we did check that the GAER tissue did significantly differentially express *Shh*, relative to the non-GAER tissue (admittedly, *Shh* was not among the top hits, likely because the GAER is only ~5 cells wide, and so makes up a relatively small proportion of the total cells of our sample – but it was still differentially expressed in the GAER region). We add a new statement to this effect from Lines 238-242.

But more importantly, in the manuscript we have spatially validated several additional factors that emerged from our analysis as differentially expressed in the GAER region of the gill arch, and we confirmed that all of these genes are, indeed, expressed in the vicinity of the GAER (as predicted by our differential expression analysis). We feel that these observations quite strongly support the accuracy of our dissections. In this study, we used our differential gene expression analysis to guide us to signalling pathways that might be interacting with GAER Shh signalling to polarise skate gill arch appendages – this led us to the Wnt signalling pathway, and we then opted to focus the result of our spatial and experimental work that pathway. And everything that we observed is consistent with activity of this pathway within the GAER region of tissue indicated in our dissection schematic in revised Figure 2.

– Multiplexed HCR RNA in situ hybridisation, in order to fully explore co–localisation of genes expressed in the GAER and non–GAER tissues i.e. SHH and FGF8.

We no longer deal with FGF signalling in this revised manuscript. In our initial figures, we had originally performed HCR for Wnt ligands and Wnt signalling readouts with a GAER marker (either *Shh* or *Wnt7a*). We opted not to show the GAER marker, so as not to obscure the signal of the other markers. But our dashed lines in the original figures were based on actual expression of a GAER marker (i.e., they were not just approximations of the location of the GAER). But in response to reviewer feedback, we now include the signal for all markers in revised Figures 3 and 4.

– In addition, to support the transcriptomic approach, it would be an improvement to see the expression of other components of the FGF and WNT signalling pathways, such as Wnt2b, Wnt3, Wnt4, and Wnt9b, FGF receptors, 1/2/3.

We focus in this paper on the Wnt signalling pathway, and we do include expression data for *Wnt2b*, *Wnt3*, *Wnt4* and *Wnt9b* (in revised Figure 3). We no longer deal with FGF signalling in this manuscript, as our dataset relating to this was incomplete and only of peripheral relevance to the focus of our revised manuscript.

– Multiplexed HCR RNA in situ hybridisation to underpin findings from the transcriptomics, for example, gene expression of tissue–specific transcription factors would also significantly lift the paper, but it is acknowledged that tissues, time, and expense may prevent this.

Our lab has just completed a move from the UK to the USA, and our lab is not yet up and running for molecular work. Additionally, the lead author on this work is currently on maternity leave. “-Omic” datasets like the one presented in this paper offer many avenues for follow-up study. In this case, we used our differential gene expression analysis to guide us to signalling pathways that might be interacting with GAER Shh signalling to polarise skate gill arch appendages – again, this analysis led us to the Wnt signalling pathway, and we opted to specifically focus the result of our spatial and experimental work on genes encoding Wnt ligands and downstream effectors. We hope that you’ll consider these data as sufficient for a paper, and we hope that the additional data from these analyses will eventually lead to further papers on other aspects of molecular patterning of skate pharyngeal arches.

3) If available for the little skate, the authors should perform KEGG analysis and gene ontology of data from the transcriptomic data set.

Thank you for this suggestion. We have now performed this analysis on our RNAseq data and found an enrichment of Wnt signalling components in the GAER region of the gill arch. We have now included this as an addition to the Methods section (Lines 196-206), a brief statement in the Results (Lines 251-254), and as a new supplemental figure and legend (Figure 2 Supplement 1).

[Editors' note: further revisions were suggested prior to acceptance, as described below.]

The reviewers are all in agreement that the manuscript has been improved and merits acceptance based on scientific merit but there are some remaining issues that need to be addressed. These are primarily to make the paper accessible. My feeling is that Figure 1 is overly complex and, as it is primarily an introduction figure, could be improved and/or reduced and combined with Figure 2.Figure1 problems–The panels are not in the same orientation, which makes it confusing to the reader.– Figure 1A is a very stunning picture– but it does not explain the gill arch anatomy in relation to the brachial ray anatomy. It would be better to have a drawn picture I think.– Figure 1A'. Not mentioned in the figure legend.– Figure 1B– this is the Anterior–posterior polarity the paper is focused on, but to the uninitiated, what the polarity is in the figure, is unclear and isn't easy to associate with the text. There is no A–P marked in this panel. Is this figure really the same orientation as A and A'? I don't understand the orientation of these samples at all, doesn't the axis of the gill arch run medial–lateral?– Figure1C. Again a different orientation. what is lateral and what is medial?– Figure 1D – the asterisk is not defined in the figure legend (it is in the body of the text). But if the asterisk marks the SHH domain in the lateral part of the gill arch, what is purple in the medial part of the gill arch? Why is the GAER not annotated here? Why is M–L not annotated? What about the HA?Figure F – Is this a summary of the gill arch? This is not stated in the figure or the text. 'distal' is added as a term. Does distal=lateral? The text for F is incomplete "…where is promotes differentiation of cartilaginous branchial'…branchial what?Figure 2 is a much better figure. I think that Figures 1 and 2 should be combined, keeping Figure 1B, C, E–E' and all of figure2. For example, Figure 2C. is a (better) version of 1D. My main confusion now is that in Figure 2, M–L seems to be reversed from Figure 1. Is this correct?These comments also relate to Fig7 and 9 which either do not annotate A–P or have M–L in a different orientation.Reviewer 2 also comments that they are unsure about Figure 5A– the DiI labelling. I think this is Figure 7? But as the figures are not numbered we cannot be sure. please check and revise if incorrect.Other confusing points in the paper– is the gill arch the same as the pharyngeal arch? These terms are used interchangeably in the paper, for example in the first two sentences. But I am not sure if they are equivalent.

We completely agree with the outstanding issues that you have flagged up, and we have addressed these issues as follows:

1. We would like to provide the readers with an anatomical overview of skate gill skeletal anatomy, so that this is not being encountered for the first time in Figure 7. But we recognise that our original Figure 1 was overly complex. We have replaced our original Figure 1 with a new and simpler figure. The revised Figure 1 shows the skate gill skeleton in ventral and lateral view, as well as a dissected gill arch. The focus of this simpler figure is the distribution of branchial rays, and how these rays relate to the proximal gill arches. We have also oriented panels A and B so that the AP axis is consistent in both. We have attached the revised figure and legend below for your review.

2. As recommended, we have incorporated the original Figure 1C and E-E’' into Figure 2. We also now consistently indicate AP and PD axes (here, and in all subsequent figures). Branchial rays and gill arches are analogous to the tetrapod limb and girdle, and so we have opted to discuss and label the axes of these appendages in a similar way. We have attached the revised figure and legend below for your review:

We also revised the first paragraph of our Results section to accommodate these new figure panels:

“In developing skate gill arches, Shh is expressed in the GAER (Figure 2A), and branchial rays develop from a GAER-responsive domain of posterior arch mesenchyme (Gillis and Hall, 2016). To further explore the transcriptional environment of the GAER and to discover additional gene expression features that may contribute to gill arch polarity, we conducted a comparative transcriptomic and differential gene expression analysis of GAER and non-GAER regions of the first gill arch of the embryonic skate. Briefly, we manually dissected the (1) GAER and (2) non-GAER (control) regions from the first gill arch of stage (S)26 skate embryos (n=5; Figure 2B–Bi), and we performed RNA extraction, library preparation, and RNAseq analysis on these samples. For the GAER region of the first gill arch, we included the whole distal tip of the arch in our dissection to ensure that the *Shh*-expressing GAER and adjacent Shh-responsive (i.e., *Ptc2*+) tissues were captured for this analysis (Figure 2C-D). Following de novo transcriptome assembly, we tested for differential gene expression between GAER and non-GAER tissues (after first testing for differential expression of *Shh* in the GAER region, to confirm dissection accuracy). We used a false discovery rate of 5% and no log fold-change (LogFC) cut-off to generate a list of 401 genes that were upregulated in the GAER region. We sorted this list for genes encoding signalling pathway components and transcription factors with a log-fold change >2. (Figure 2E). Differentially expressed genes were subjected to functional enrichment analysis, and this analysis indicated enrichment of Wnt signalling pathway components in GAER region (Figure 2 Supplement 1 for enrichment analysis). Differentially expressed genes in the GAER region included those encoding the Wnt ligands Wnt2b, Wnt3, Wnt4, Wnt7b and Wnt9, the transmembrane Wnt inhibitors Kremen1 (Mao et al., 2002) and APCDD1 (Shimomura et al., 2010), and the secreted Wnt antagonist Notum (Zhang et al., 2015). We therefore chose to further investigate involvement of the Wnt pathway in GAER signalling by spatially validating the expression of these genes, using the most highly expressed isoforms for mRNA *in situ* hybridization by chain reaction (HCR) probe set design.”

3. We have reoriented the panels in Figures 7 and 8 so that anterior is always to the left, and (where relevant) we have indicated the AP and PD axes with labels in Figures 7, 8 and 9.

4. There were a few incorrect figure references in the text (a holdover from the figure numbering in our original submission).

We have gone through the text and corrected these.

5. “Other confusing points in the paper- is the gill arch the same as the pharyngeal arch? These terms are used interchangeably in the paper, for example in the first two sentences. But I am not sure if they are equivalent.”

Vertebrate pharyngeal arches are a series of paired structures that form on either side of the embryonic head. These arches are named according to their ancestral skeletal derivatives: The first of these arches is called the mandibular arch, the second is called the hyoid arch, and the caudal pharyngeal arches are collectively called the gill arches. Unfortunately, these skeletal derivatives of pharyngeal arches are also generally referred to as “arches”, and this can lead to some confusion when referring to embryonic precursors and their derivatives. To try and clarify this, we have revised the first paragraph of our introduction as follows:

“The pharyngeal arches of vertebrates are a series of paired columns of tissue that form on either side of the embryonic head. Pharyngeal arches form as iterative outpockets of foregut endoderm contact the overlying surface ectoderm, and this meeting of endoderm and ectoderm generates columns lined laterally by ectoderm, medially by endoderm, and containing a core of mesoderm and neural crest-derived mesenchyme (Graham and Smith, 2001). In fishes, endodermal outpockets fuse with the overlying surface ectoderm, giving rise to the gill slits and the respiratory surfaces of the gills (Gillis and Tidswell, 2017). In amniotes, endodermal outpockets give rise to glandular tissues, such as the tonsils, parathyroid and ultimobranchial glands (Grevellec and Tucker, 2010). The largely neural crest-derived mesenchyme of the pharyngeal arches gives rise to the pharyngeal skeleton – i.e., the skeleton of the jaws and gills in fishes, and of the jaw, auditory ossicles, and larynx in amniotes (Jiang et al., 2002, Kague et al., 2012, Sleight and Gillis, 2020, Couly and Le Douarin, 1990) – with mesenchymal derivatives of each arch receiving patterning and polarity information via signals from adjacent epithelia (Veitch et al., 1999, Gillis et al., 2009a, Couly et al., 2002, Brito et al., 2006). Pharyngeal arches are named according to their ancestral skeletal derivatives: the 1^st^ pharyngeal arch is termed the mandibular arch, the 2^nd^ pharyngeal arch is the hyoid arch, while the caudal pharyngeal arches are collectively termed the gill arches. In fishes, the gill arches give rise to a series of skeletal “arches” that support the respiratory lamellae of the gills.”